# Beyond Concept Bottleneck Models:
# How to Make Black Boxes Intervenable?

**Sonia Laguna,** * **Ričards Marcinkevičs,** * **Moritz Vandenhirtz, Julia E. Vogt**
Department of Computer Science, ETH Zurich, Switzerland

## Abstract

Recently, interpretable machine learning has re-explored concept bottleneck models (CBM). An advantage of this model class is the user's ability to intervene on predicted concept values, affecting the downstream output. In this work, we introduce a method to perform such concept-based interventions on *pretrained* neural networks, which are not interpretable by design, only given a small validation set with concept labels. Furthermore, we formalise the notion of *intervenability* as a measure of the effectiveness of concept-based interventions and leverage this definition to fine-tune black boxes. Empirically, we explore the intervenability of black-box classifiers on synthetic tabular and natural image benchmarks. We focus on backbone architectures of varying complexity, from simple, fully connected neural nets to Stable Diffusion. We demonstrate that the proposed fine-tuning improves intervention effectiveness and often yields better-calibrated predictions. To showcase the practical utility of our techniques, we apply them to deep chest X-ray classifiers and show that fine-tuned black boxes are more intervenable than CBMs. Lastly, we establish that our methods are still effective under vision-language-model-based concept annotations, alleviating the need for a human-annotated validation set.

## 1 Introduction

Interpretable and explainable machine learning (Doshi-Velez & Kim, 2017; Molnar, 2022) have seen a renewed interest in concept-based predictive models and approaches to post hoc explanation, such as concept bottlenecks (Lampert et al., 2009; Kumar et al., 2009; Koh et al., 2020), contextual semantic interpretable bottlenecks (Marcos et al., 2020), concept whitening layers (Chen et al., 2020), and concept activation vectors (B. Kim et al., 2018). Moving beyond interpretations defined in the high-dimensional, unwieldy input space, these techniques relate the model's inputs and outputs via additional high-level human-understandable attributes, also referred to as *concepts*. Typically, neural network models are supervised to predict these attributes in a dedicated bottleneck layer, or post hoc explanations are derived to measure the model's sensitivity to concept variables.

This work focuses specifically on the concept bottleneck models, as revisited by Koh et al. (2020). In brief, a CBM is a neural network consisting of successive concept and target prediction modules, where the final output depends on the input solely through the predicted concepts. Such models are trained on labelled data, in addition, annotated by attributes. At inference time, a human user may interact with the CBM by editing the predicted concept values, which, as a result, affects the downstream target prediction. This act of model editing is known as an *intervention*. The user's ability to intervene is a compelling advantage of CBMs over other interpretable model classes, in that the former allows for human-model interaction.

---

*Equal contribution. Correspondence to `sonia.lagunacillero@inf.ethz.ch`

38th Conference on Neural Information Processing Systems (NeurIPS 2024).

In contrast to previous works (Yuksekgonul et al., 2023; Oikarinen et al., 2023), we focus on *instance-specific* interventions, *i.e.* performed individually for each data point. To this end, we explore two questions: **(i)** *Given a small validation set with concept labels, how can we perform instance-specific interventions directly on a pretrained black-box model?* **(ii)** *How can we fine-tune the black-box model to improve the effectiveness of interventions performed on it?*

Such instance-specific interventions can be relevant in high-stakes decisions. Our specific motivation is healthcare. For instance, consider computer-aided diagnosis, where a doctor may make decisions assisted by a predictive model. In this setting, the doctor handles patients on a *case-by-case* basis and may benefit from instance-specific interactions with the black box. While, in principle, a specialist may just override predictions, in many cases, concept and target variables are linked via nonlinear relationships potentially unknown to the user. Figure 1 contains a simplified, intuitive example of an instance-specific concept-based intervention for natural images. Additional and more comprehensive examples can be found in Appendix A.

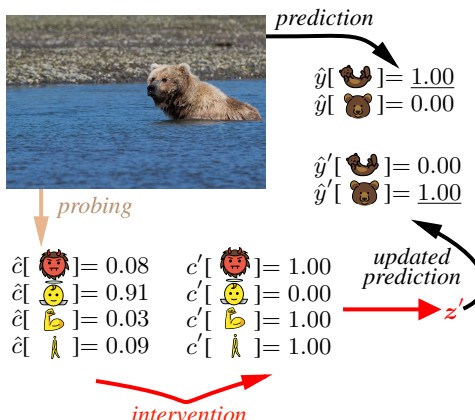

Figure 1: A simplified, intuitive example: an image of a grizzly bear is wrongly identified as an otter. Our method allows performing a concept-based intervention and flip the predicted class. In order of appearance from left to right and top to bottom, the depicted concepts and classes are "fierce", "timid", "muscle", "walks", "otter", and "grizzly bear".

**Contributions** This work contributes to the research on concept bottleneck models and concept-based explanations. **(1)** We devise a simple procedure that, given a set of concepts and a small labelled validation set, allows performing concept-based instance-specific interventions (Figure 1) on a pretrained black-box neural network by editing its activations at an intermediate layer. Notably, concept labels are not required in the large *training set*, and the network's architecture does not need to be adjusted. **(2)** We formalise *intervenability* as a measure of the effectiveness of interventions performed on the model. Utilising intervenability as a loss, we introduce a novel fine-tuning procedure. This fine-tuning strategy is designed to improve the effectiveness of concept-based interventions. It preserves the original model's architecture and representations to be used in downstream tasks. **(3)** We evaluate the proposed procedures alongside several baselines on the synthetic tabular, natural image, and medical imaging data. We demonstrate that in practice, for studied classification problems, we can improve the predictive performance of pretrained black-box models via concept-based interventions. We investigate fully connected and more complex backbone architectures. We show that the effectiveness of interventions improves considerably when explicitly fine-tuning for intervenability. Lastly, we observe that our methods are successful in datasets where concept labels are acquired using vision-language models (VLM), alleviating the need for a human annotation.

## 2  Related Work

The use of high-level attributes in predictive models has been well-explored in computer vision (Lampert et al., 2009; Kumar et al., 2009). Recent efforts have focused on explicitly incorporating concepts in neural networks (Koh et al., 2020; Marcos et al., 2020), producing high-level post hoc explanations by quantifying the network's sensitivity to the attributes (B. Kim et al., 2018), probing (Alain & Bengio, 2016; Belinkov, 2022) and de-correlating and aligning the network's latent space with concept variables (Chen et al., 2020). Other works (Xie et al., 2020) have studied the use of auxiliary external attributes in out-of-distribution settings. To alleviate the assumption of being given interpretable concepts, some have explored concept discovery prior to post hoc explanation (Ghorbani et al., 2019; Yeh et al., 2020). Another relevant line of work investigated concept-based counterfactual explanations (CCE) (Abid et al., 2022; S. Kim et al., 2023).

Concept bottleneck models (Koh et al., 2020) have sparked a renewed interest in concept-based classification methods. Many related works have described the inherent limitations of this model class and attempted to address them (Margeloiu et al., 2021; Mahinpei et al., 2021; Marconato et al.,

2022; Havasi et al., 2022; Sawada & Nakamura, 2022; Marcinkevičs et al., 2024). Another line of research has investigated modelling uncertainty and probabilistic extensions of the CBMs (Collins et al., 2023; E. Kim et al., 2023). Most related to the current paper are the techniques for converting pretrained black-box neural networks into CBMs post hoc (Yuksekgonul et al., 2023; Oikarinen et al., 2023) by keeping the network's backbone and projecting its activations into the concept space. Additionally, these works explore automated concept discovery using VLMs.

As mentioned, CBMs allow for concept-based instance-specific interventions. Several follow-up works have studied interventions in further detail. Chauhan et al. (2023) and Sheth et al. (2022) introduce adaptive intervention policies to further improve the predictive performance of the CBMs at the test time. In a similar vein, Steinmann et al. (2023) propose learning to detect mistakes in the predicted concepts and, thus, learning intervention strategies. Shin et al. (2023) empirically investigate different intervention procedures across various settings.

## 3    Methods

In this section, we define a measure for the effectiveness of concept-based interventions and present a technique for intervening on black-box neural networks. Furthermore, we propose a fine-tuning procedure to improve the effectiveness of such interventions. Additional remarks beyond the current scope are included in Appendix C.

In the remainder of this paper, we will adhere to the following notation. Let $x \in \mathcal{X}$, $y \in \mathcal{Y}$, and $c \in \mathcal{C}$ be the covariates, targets, and concepts. A CBM $f_{\theta}$, parameterised by $\theta$, is given by $f_{\theta}(x) = g_{\psi}(h_{\phi}(x))$, where $h_{\phi} : \mathcal{X} \to \mathcal{C}$ maps inputs to predicted concepts, *i.e.* $\hat{c} = h_{\phi}(x)$, and $g_{\psi} : \mathcal{C} \to \mathcal{Y}$ predicts the target based on $\hat{c}$, *i.e.* $\hat{y} = g_{\psi}(\hat{c})$. CBMs are trained on data points $(x, c, y)$ and are supervised by the concept and target prediction losses. At test time, if the user chooses to replace $\hat{c}$ with another $c' \in \mathcal{C}$, *i.e.* intervene, the final prediction is given by $\hat{y}' = g_{\psi}(c')$.

Next to CBMs, we will consider a black-box neural network $f_{\theta} : \mathcal{X} \to \mathcal{Y}$ parameterised by $\theta$ and a slice $\langle g_{\psi}, h_{\phi} \rangle$ (Leino et al., 2018), defining a layer, s.t. $f_{\theta}(x) = g_{\psi}(h_{\phi}(x))$. We will assume that the black box has been trained end-to-end on the labelled data $\{(x_i, y_i)\}_i$. Lastly, for all techniques, we will assume being given a small *validation* set $\{(x_i, c_i, y_i)\}_i$.

### 3.1    Intervening on Black-box Models

Given a black-box model $f_{\theta}$ and a data point $(x, y)$, a human user might desire to influence the prediction $\hat{y} = f_{\theta}(x)$ made by the model via high-level and understandable concept values $c'$, *e.g.* think of a doctor trying to interact with a chest X-ray classifier ($f_{\theta}$) by annotating their findings ($c'$) in a radiograph ($x$), where findings correspond to the clinical concepts, such as the presence of edema or fracture. To facilitate such interactions, we propose a simple recipe for concept-based instance-specific interventions (detailed in Figure 2) that can be applied to *any* black-box neural network model. Intuitively, using the given validation data and concept values, our procedure edits the network's representations $z = h_{\phi}(x)$, where $z \in \mathcal{Z}$, to align more closely with $c'$ and, thus, affects the downstream prediction. Below, we explain this procedure step-by-step. Pseudocode implementation can be found as part of Algorithm B.1 in Appendix B.

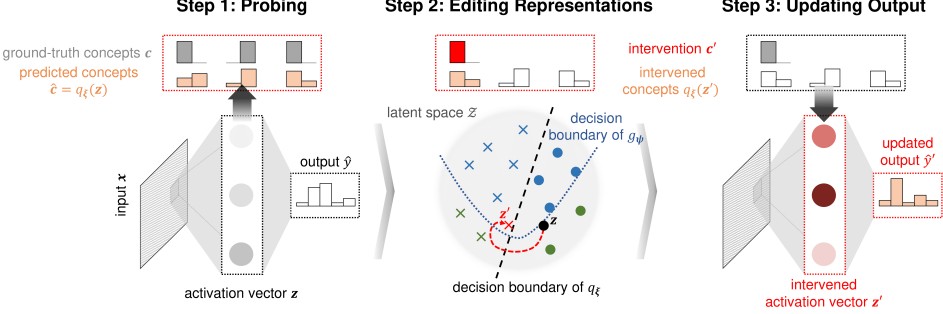

Figure 2: Three steps of the intervention procedure. (i) A probe $q_{\xi}$ is trained to predict the concepts $c$ from the activation vector $z$. (ii) The representations are edited according to Equation 1. (iii) The final prediction is updated to $\hat{y}'$ based on the edited representations $z'$.

**Step 1: Probing** To align the network's activation vectors with concepts, the preliminary step is to train a probing function (Alain & Bengio, 2016; B. Kim et al., 2018; Belinkov, 2022), or a *probe* for short, mapping the intermediate representations to concepts. Namely, using the given annotated validation data $\{(\boldsymbol{x}_i, \boldsymbol{c}_i, y_i)\}_i$, we train a multivariate probe $q_{\boldsymbol{\xi}}$ to predict the concepts $\boldsymbol{c}_i$ from the representations $\boldsymbol{z}_i = h_{\boldsymbol{\phi}}(\boldsymbol{x}_i)$: $\min_{\boldsymbol{\xi}} \sum_i \mathcal{L}^c(q_{\boldsymbol{\xi}}(\boldsymbol{z}_i), \boldsymbol{c}_i)$, where $\mathcal{L}^c$ is the concept prediction loss. Note that, herein, an essential design choice explored in our experiments is the (non)linearity of the probe. Consequently, the probing function can be used to interpret the activations in the intermediate layer and edit them.

**Step 2: Editing Representations** Recall that we are given a data point $(\boldsymbol{x}, y)$ and concept values $\boldsymbol{c}'$ for which an intervention needs to be performed. Note that this $\boldsymbol{c}' \in \mathcal{C}$ could correspond to the ground-truth concept values or reflect the beliefs of the human subject intervening on the model. Intuitively, we seek an activation vector $\boldsymbol{z}'$, which is similar to $\boldsymbol{z} = h_{\boldsymbol{\phi}}(\boldsymbol{x})$ and consistent with $\boldsymbol{c}'$ according to the previously learnt probing function $q_{\boldsymbol{\xi}}$: $\arg\min_{\boldsymbol{z}'} d(\boldsymbol{z}, \boldsymbol{z}')$, s.t. $q_{\boldsymbol{\xi}}(\boldsymbol{z}') = \boldsymbol{c}'$, where $d$ is an appropriate distance function applied to the activation vectors from the intermediate layer. Throughout main experiments (Section 4), we utilise the Euclidean distance, which is frequently applied to neural network representations, *e.g.* see works by Moradi Fard et al. (2020) and Jia et al. (2021). In Appendix F.8, we additionally explore the cosine distance. Instead of the constrained problem above, we resort to minimising a relaxed objective:

$$\arg\min_{\boldsymbol{z}'} \lambda \mathcal{L}^c(q_{\boldsymbol{\xi}}(\boldsymbol{z}'), \boldsymbol{c}') + d(\boldsymbol{z}, \boldsymbol{z}'), \tag{1}$$

where, similarly to the counterfactual explanations (Wachter et al., 2017; Mothilal et al., 2020), hyperparameter $\lambda > 0$ controls the tradeoff between the intervention's validity, *i.e.* the "consistency" of $\boldsymbol{z}'$ with the given concept values $\boldsymbol{c}'$ according to the probe, and proximity to the original activation vector $\boldsymbol{z}$. In practice, we optimise $\boldsymbol{z}'$ for batched interventions using Adam (Kingma & Ba, 2015). Appendix F.2 explores the effect of $\lambda$ on the post-intervention distribution of representations.

**Step 3: Updating Output** The edited $\boldsymbol{z}'$ can be consequently fed into $g_{\boldsymbol{\psi}}$ to compute the updated output $\hat{y}' = g_{\boldsymbol{\psi}}(\boldsymbol{z}')$, which could be then returned and displayed to the human subject. For example, if $\boldsymbol{c}'$ are the ground-truth concept values, we would ideally expect a decrease in the prediction error for the given data point $(\boldsymbol{x}, y)$.

### 3.2 What is Intervenability?

Concept bottlenecks (Koh et al., 2020) and their extensions are often evaluated empirically by plotting test-set performance or error attained after intervening on concept subsets of varying sizes. Ideally, the model's test-set performance should improve when given more ground-truth attribute values. Below, we formalise this notion of intervention effectiveness, referred to as *intervenability*, for the concept bottleneck and black-box models.

For a trained CBM $f_{\boldsymbol{\theta}}(\boldsymbol{x}) = g_{\boldsymbol{\psi}}(h_{\boldsymbol{\phi}}(\boldsymbol{x})) = g_{\boldsymbol{\psi}}(\hat{\boldsymbol{c}})$, where $\hat{\boldsymbol{c}}$ are the predicted concept values, we define the intervenability as follows:

$$\mathbb{E}_{(\boldsymbol{x},\boldsymbol{c},y)\sim\mathcal{D}}\left[\mathbb{E}_{\boldsymbol{c}'\sim\pi}\left[\mathcal{L}^y\Big(\underbrace{f_{\boldsymbol{\theta}}(\boldsymbol{x})}_{\hat{y}=g_{\boldsymbol{\psi}}(\hat{\boldsymbol{c}})}, y\Big) - \mathcal{L}^y\Big(\underbrace{g_{\boldsymbol{\psi}}(\boldsymbol{c}')}_{\hat{y}'}, y\Big)\right]\right], \tag{2}$$

where $\mathcal{D}$ is the joint distribution over the covariates $\boldsymbol{x}$, concepts $\boldsymbol{c}$, and targets $y$, $\mathcal{L}^y$ is the target prediction loss, *e.g.* the mean squared error (MSE) or cross-entropy (CE), and $\pi$ denotes a distribution over edited concept values $\boldsymbol{c}'$. Observe that Equation 2 generalises the standard evaluation strategy of intervening on a random concept subset and setting it to the ground-truth values, as proposed in the original work by Koh et al. (2020). Here, the effectiveness of interventions is quantified by the gap between the regular prediction loss and the loss attained after the intervention: the larger the gap between these values, the stronger the effect interventions have. The intervenability measure is loosely related to permutation-based variable importance and model reliance (Fisher et al., 2019). We provide a discussion of this relationship in Appendix C.

Note that the definition in Equation 2 can also accommodate more sophisticated intervention strategies, for example, similar to those studied by Shin et al. (2023) and Sheth et al. (2022). An intervention strategy can be specified via the distribution $\pi$, which can be conditioned on $\boldsymbol{x}$, $\hat{\boldsymbol{c}}$, $\boldsymbol{c}$, $\hat{y}$,

or even $y$: $\pi\left(c' | x, \hat{c}, c, \hat{y}, y\right)$. The set of conditioning variables may vary across application scenarios. For brevity, we will use $\pi$ as a shorthand notation for this distribution. Lastly, notice that, in practice, when performing human- or application-grounded evaluation (Doshi-Velez & Kim, 2017), sampling from $\pi$ may be replaced with the interventions by a human. Algorithms E.1 and E.2 provide concrete examples of the strategies utilised in our experiments.

Leveraging the intervention procedure described in Section 3.1, analogous to Equation 2, the intervenability for a black-box neural network $f_{\boldsymbol{\theta}}$ at the intermediate layer given by $\langle g_{\boldsymbol{\psi}}, h_{\boldsymbol{\phi}} \rangle$ is

$$
\begin{aligned}
&\mathbb{E}_{(x,c,y)\sim\mathcal{D},\, c'\sim\pi}\left[\mathcal{L}^y\left(f_{\boldsymbol{\theta}}\left(x\right), y\right) - \mathcal{L}^y\left(g_{\boldsymbol{\psi}}\left(z'\right), y\right)\right], \\
&\text{where } z' \in \arg\min_{\tilde{z}} \lambda \mathcal{L}^c\left(q_{\boldsymbol{\xi}}\left(\tilde{z}\right), c'\right) + d\left(z, \tilde{z}\right).
\end{aligned}
\tag{3}
$$

Recall that $q_{\boldsymbol{\xi}}$ is the probe trained to predict $c$ based on the activations $h_{\boldsymbol{\phi}}\left(x\right)$ (step 1, Section 3.1). Furthermore, in the first line of Equation 3, edited representations $z'$ are a function of $c'$, as defined by the second line, which corresponds to step 2 of the intervention procedure (Equation 1).

### 3.3 Fine-tuning for Intervenability

Since the intervenability measure defined in Equation 3 is differentiable, a neural network can be fine-tuned by explicitly maximising it using, for example, mini-batch gradient descent. We expect fine-tuning for intervenability to reinforce the model's reliance on the high-level attributes and have a regularising effect. In this section, we provide a detailed description of the fine-tuning procedure (Algorithm B.1, Appendix B), and, afterwards, we demonstrate its practical utility empirically.

Naïvely optimising intervenability from Equation 3 may decrease the predictive performance. Therefore, to fine-tune an already trained black-box model $f_{\boldsymbol{\theta}}$, we combine the intervenability term with the target prediction loss, which amounts to the following optimisation problem:

$$
\min_{\boldsymbol{\psi}, z'} \mathbb{E}_{(x,c,y)\sim\mathcal{D},\, c'\sim\pi}\left[\mathcal{L}^y\left(g_{\boldsymbol{\psi}}\left(z'\right), y\right)\right], \text{ s.t. } z' \in \arg\min_{\tilde{z}} \lambda \mathcal{L}^c\left(q_{\boldsymbol{\xi}}\left(\tilde{z}\right), c'\right) + d\left(z, \tilde{z}\right). \tag{4}
$$

Notably, Equation 4 can be generalised by introducing a weight for the intervenability term:

$$
\begin{aligned}
&\min_{\boldsymbol{\phi}, \boldsymbol{\psi}, z'} \mathbb{E}_{(x,c,y)\sim\mathcal{D},\, c'\sim\pi}\left[(1-\beta)\,\mathcal{L}^y\left(g_{\boldsymbol{\psi}}\left(h_{\boldsymbol{\phi}}\left(x\right)\right), y\right) + \beta\mathcal{L}^y\left(g_{\boldsymbol{\psi}}\left(z'\right), y\right)\right], \\
&\text{s.t. } z' \in \arg\min_{\tilde{z}} \lambda \mathcal{L}^c\left(q_{\boldsymbol{\xi}}\left(\tilde{z}\right), c'\right) + d\left(z, \tilde{z}\right),
\end{aligned}
\tag{5}
$$

where $\beta \in (0, 1]$ is the aforementioned weight. Note that for $\beta = 1$, the optimisation simplifies to Equation 4. For simplicity, we treat the probe's parameters $\boldsymbol{\xi}$ as fixed. However, since the outer optimisation problem is defined w.r.t. $\boldsymbol{\phi}$, ideally, the probe would need to be optimised as the third, inner-most level. By contrast, in the simplified setting under $\beta = 1$ (Equation 4), the parameters of $h_{\boldsymbol{\phi}}$ do not need to be optimised, and, hence, the probing function can be left fixed, as activations $z$ are not affected by the fine-tuning. We consider this case to (i) computationally simplify the problem, avoiding trilevel optimisation, and (ii) keep the network's representations unchanged after fine-tuning for purposes of transfer learning for other downstream tasks. In practice, fine-tuning is performed by intervening on batches of data points. Since interventions can be executed online using a GPU (within seconds), our approach is computationally feasible.

## 4 Experimental Setup

**Datasets** We evaluate the proposed methods on synthetic and real-world benchmarks summarised in Table D.1 (Appendix D). Across all experiments, fine-tuning has been performed exclusively on the validation data, and evaluation—on the test set. Further details can be found in Appendix D.

For controlled experiments, we have adapted the nonlinear **synthetic** tabular dataset from Marcinkevičs et al. (2024). We consider two data-generating mechanisms shown in Figure D.1 (Appendix D.1): *bottleneck*, and *incomplete*. The first scenario directly matches the inference graph of the vanilla CBM. The *incomplete* is a scenario with incomplete concepts, where $c$ does not fully explain the association between $x$ and $y$, with unexplained variance modelled via a residual connection.

Another benchmark we consider is the **Animals with Attributes 2 (AwA2)** natural image dataset (Lampert et al., 2009; Xian et al., 2019). It includes animal images accompanied by 85 binary attributes and species labels. To further corroborate our findings, we perform experiments on the Caltech-UCSD Birds-200-2011 (**CUB**) dataset (Wah et al., 2011) (Appendix D.3), adapted for the CBM setting as described by Koh et al. (2020). We report the CUB results in Appendix F.4.

To investigate settings *without* human-annotated concept values, we evaluate our method on **CIFAR-10** (Krizhevsky et al., 2009) and the large-scale **ImageNet** (Russakovsky et al., 2015) natural image datasets. Following the previous literature (Oikarinen et al., 2023), we use concepts generated by GPT-3. Concept labels are produced based on CLIP (Radford et al., 2021) similarities between each image and verbal descriptions. We utilise 143 attributes for CIFAR-10 and 100 for ImageNet. ImageNet results are reported in Appendix F.5.

Finally, we explore a practical setting of chest radiograph classification. Namely, we test the techniques on public **MIMIC-CXR** (Johnson et al., 2019) and **CheXpert** (Irvin et al., 2019) datasets from the Beth Israel Deaconess Medical Center, Boston, MA, and Stanford Hospital. Both datasets have 14 binary attributes extracted from radiologist reports. In our analysis, the *Finding/No Finding* attribute is the target variable, and the remaining labels are the concepts, similar to Chauhan et al. (2023). For simplicity, we retain a single X-ray per patient, excluding data with uncertain labels. The results on CheXpert are similar to those on MIMIC-CXR and can be found in Appendix F.6.

**Baselines & Methods**  Below, we briefly outline the neural network models and fine-tuning techniques compared. All methods were implemented using PyTorch (v 1.12.1) (Paszke et al., 2019). Appendix E provides additional details. The code is available in a repository at `https://github.com/sonialagunac/Beyond-CBM`.

Firstly, we train a standard neural network (**BLACK BOX**) without concept knowledge, *i.e.* on the dataset of tuples $\{(\boldsymbol{x}_i, y_i)\}_i$. We utilise our technique for intervening post hoc by training a probe to predict concepts and editing the network's activations (Equation 1, Section 3.1). All experiments reported in Section 5 use a linear probe, while the nonlinearity is explored in Appendix F. As an interpretable baseline, we consider the vanilla concept bottleneck model (**CBM**) by Koh et al. (2020). Across all experiments, we restrict ourselves to the joint bottleneck version, which minimises the weighted sum of the target and concept prediction losses: $\min_{\boldsymbol{\phi},\boldsymbol{\psi}} \mathbb{E}_{(\boldsymbol{x},\boldsymbol{c},y)\sim\mathcal{D}} \left[\mathcal{L}^y\left(f_{\boldsymbol{\theta}}\left(\boldsymbol{x}\right), y\right) + \alpha\mathcal{L}^c\left(h_{\boldsymbol{\phi}}\left(\boldsymbol{x}\right), \boldsymbol{c}\right)\right]$, where $\alpha > 0$ is a hyperparameter controlling the tradeoff between the two loss terms. Finally, as the primary method of interest, we apply our fine-tuning for intervenability technique (**FINE-TUNED, I**; Equation 4, Section 3.3) on the annotated validation set $\{(\boldsymbol{x}_i, \boldsymbol{c}_i, y_i)\}_i$.

In addition, as a common-sense baseline, we fine-tune the black box by training a probe to predict the concepts from intermediate representations (**FINE-TUNED, MT**). This amounts to multitask (MT) learning with hard weight sharing (Ruder, 2017). Specifically, the model is fine-tuned by minimising the following MT loss: $\min_{\boldsymbol{\phi},\boldsymbol{\psi},\boldsymbol{\xi}} \mathbb{E}_{(\boldsymbol{x},\boldsymbol{c},y)\sim\mathcal{D}}[\mathcal{L}^y\left(f_{\boldsymbol{\theta}}\left(\boldsymbol{x}\right), y\right) + \alpha\mathcal{L}^c\left(q_{\boldsymbol{\xi}}\left(h_{\boldsymbol{\phi}}\left(\boldsymbol{x}\right)\right), \boldsymbol{c}\right)]$. Interventions on this model are performed using the three-step approach introduced in Section 3.1.

As another baseline, we fine-tune the black box by appending concepts to the network's activations (**FINE-TUNED, A**). At test time, unknown concept values are set to 0.5. To prevent overfitting and handle missingness, randomly chosen concept variables are masked during training. The objective is given by $\min_{\tilde{\boldsymbol{\psi}}} \mathbb{E}_{(\boldsymbol{x},\boldsymbol{c},y)\sim\mathcal{D}}[\mathcal{L}^y(\tilde{g}_{\tilde{\boldsymbol{\psi}}}\left(\left[h_{\boldsymbol{\phi}}\left(\boldsymbol{x}\right), \boldsymbol{c}\right]\right), y)]$, where $\tilde{g}$ takes as input concatenated activation and concept vectors. Note that, for this baseline, the parameters $\boldsymbol{\phi}$ remain fixed during fine-tuning.

Last but not least, as a strong baseline resembling the approaches by Yuksekgonul et al. (2023) and Oikarinen et al. (2023), we train a CBM post hoc (**POST HOC CBM**) using *sequential* optimisation. Our implementation follows the original methods by Yuksekgonul et al. (2023) and Oikarinen et al. (2023), while adjusting some design choices to make the techniques more readily comparable. The optimisation comprises two steps: (i) $\hat{\boldsymbol{\xi}} = \arg\min_{\boldsymbol{\xi}} \mathbb{E}_{(\boldsymbol{x},\boldsymbol{c},y)\sim\mathcal{D}}[\mathcal{L}^c\left(q_{\boldsymbol{\xi}}\left(h_{\boldsymbol{\phi}}\left(\boldsymbol{x}\right)\right), \boldsymbol{c}\right)]$, (ii) $\min_{\boldsymbol{\psi}} \mathbb{E}_{(\boldsymbol{x},\boldsymbol{c},y)\sim\mathcal{D}}[\mathcal{L}^y(g_{\boldsymbol{\psi}}(q_{\hat{\boldsymbol{\xi}}}(h_{\boldsymbol{\phi}}(\boldsymbol{x}))), y)]$. Additionally, we explore the impact of residual modelling (Yuksekgonul et al., 2023) in Appendix F.9. The architectures of individual modules were kept as similar as possible for a fair comparison across all techniques.

**Evaluation** To compare the methods, we conduct interventions and analyse model performance under varying concept subset sizes. We report the areas under the receiver operating characteristic (AUROC) and precision-recall curves (AUPR) (Davis & Goadrich, 2006) since these performance measures provide a well-rounded summary over varying cutoff points and it might be challenging to choose a single cutoff in high-stakes decision areas. We utilise the Brier score (Brier, 1950) to gauge the accuracy of probabilistic predictions and, in addition, evaluate calibration.

## 5 Results

**Results on Synthetic Data** Figures 3(f) and 4(a) show intervention results obtained across ten independent simulations under the two generative mechanisms (Figure D.1, Appendix D.1) on the synthetic tabular data. We observe that, in principle, the proposed intervention procedure can improve the predictive performance of a black-box neural network. However, in the bottleneck scenario, interventions are considerably more effective in CBMs than in untuned black-box classifiers since the underlying generative process directly matches the CBM's architecture. Models explicitly fine-tuned for intervenability (FINE-TUNED, I) significantly improve over the original classifier, achieving intervention curves comparable to those of the CBM.

Importantly, under an *incomplete* concept set, black-box classifiers are superior to the ante hoc CBM because not all concepts relevant to the target prediction are given. Moreover, fine-tuning for intervenability improves intervention effectiveness while maintaining the performance gap. This experiment suggests the superiority of our method in settings where the concept set does not capture all label-relevant information. Other fine-tuning strategies (FINE-TUNED, MT and FINE-TUNED, A) are either less effective or harmful, leading to a lower increase in AUROC and AUPR than attained by the untuned black box. Lastly, CBMs trained post hoc perform well in the simple *bottleneck* scenario, being only slightly less intervenable than FINE-TUNED, I. However, for the *incomplete* setting, interventions hurt the performance of the post hoc CBM. This behaviour may be related to the leakage (Havasi et al., 2022) and is not mitigated by residual modelling explored in Appendix F.9.

To study the influence of the validation set size ($N_{val}$) on probing and fine-tuning, we perform ablations under the *bottleneck* scenario (Figure 3). For a fair comparison w.r.t. sample efficiency, here, we train a CBM on the dataset of the *same* size. While the effectiveness of interventions on FINE-TUNED, I is slightly hampered by smaller validation sets, the decrease is moderate. We observe impactful interventions with validation set sizes as small as 0.5% of the original one (Figure 3(a)). Across all settings, our method remains superior to baselines. Importantly, our fine-tuning approach has a

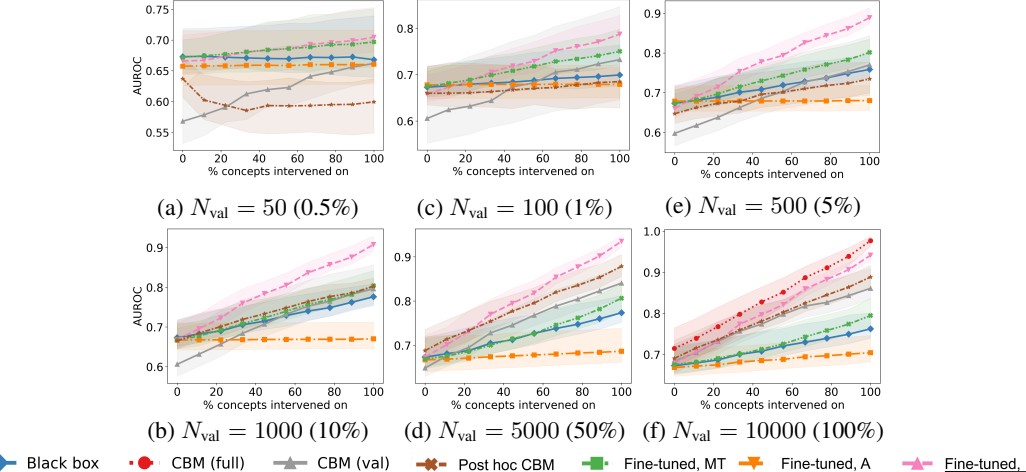

Figure 3: Intervention results w.r.t. target AUROC on the synthetic *bottleneck* data. We explore the performance under varying validation set sizes ($N_{val}$). Percentages correspond to the fractions of the *original* validation set. For CBMs, we report the results obtained by training on the validation (**CBM val**) and full training sets (**CBM full**). Interventions were performed on test data across ten simulations. Lines correspond to medians, and confidence bands are given by interquartile ranges.

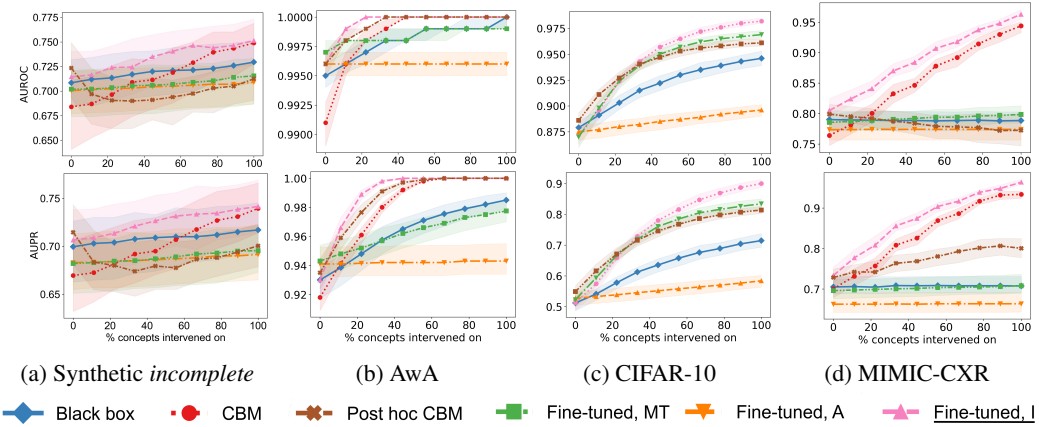

(a) Synthetic *incomplete*  (b) AwA  (c) CIFAR-10  (d) MIMIC-CXR

Black box    CBM    Post hoc CBM    Fine-tuned, MT    Fine-tuned, A    Fine-tuned, I

Figure 4: Intervention results on the (a) synthetic *incomplete*, (b) AwA2, (c) CIFAR-10, and (d) MIMIC-CXR datasets w.r.t. target AUROC (*top*) and AUPR (*bottom*) across ten seeds.

better sample efficiency than post hoc CBM, which exhibits a considerable dropoff in intervention effectiveness. Likewise, the performance of the CBMs decreases, suggesting their limited utility under smaller dataset sizes. Analogous results w.r.t. AUPR are reported in Appendix F.1.

In Table 1, we report the test-set performance of the models *without* interventions (under the *bottleneck* mechanism). For the concept prediction, CBM outperforms black-box models, even after fine-tuning with the MT loss. However, without interventions, all models attain comparable AUROCs and AUPRs at the target prediction. Interestingly, FINE-TUNED, I results in better-calibrated probabilistic predictions with lower Brier scores than those made by the original black box and after applying other fine-tuning strategies. As evidenced by Figure F.9(a) (Appendix F.7), fine-tuning has a regularising effect, reducing the false overconfidence observed in neural networks (Guo et al., 2017).

In the supplementary material, we report several additional findings. Figure F.2 (Appendix F.1) contains further ablations for the intervention procedure on the influence of the hyperparameters, intervention strategies, and probe. In addition, Appendix F.2 explores the effect of interventions on the distribution of representations. Lastly, in Appendix F.10, we show that the performance of the CBM on this dataset is not sensitive to the choice of the optimisation procedure (Koh et al., 2020).

**Results on AwA2**    Additionally, we explore the AwA2 dataset in Figure 4(b). This is a simple classification benchmark with class-wide concepts helpful for predicting the target. Hence, CBMs trained ante and post hoc are highly performant and intervenable. Nevertheless, untuned black-box models also benefit from concept-based interventions. In agreement with our findings on the synthetic dataset and in contrast to the other fine-tuning methods, ours enhances the performance of black-box models. Notably, black boxes fine-tuned for intervenability even surpass CBMs. Overall, the simplicity of this dataset leads to the generally high AUROCs and AUPRs across all methods.

To further investigate the impact of hyperparameters on the interventions, we have performed ablation studies on untuned black boxes. These results are reported in Figures F.4(a)–(c), Appendix F.3. In brief, we observe that interventions are effective across all values of the $\lambda$-parameter from Equation 3 (Figure F.5(a)). Expectedly, higher values yield a steeper increase in AUROC and AUPR. Figure F.4(b) compares two intervention strategies: randomly selecting concepts (random) and prioritising the most uncertain ones (uncertainty) (Shin et al., 2023) to intervene on (Algorithms E.1 and E.2, Appendix E). The strategy has an impact on the performance increase, with the uncertainty-based approach yielding a steeper improvement. Finally, Figure F.4(c) compares linear and nonlinear probes. Here, intervening via a nonlinear function leads to a higher performance increase.

To show the efficacy of our methods across different backbone architectures, in Appendix F.3, we also explore AwA2 with the Inception (Szegedy et al., 2015) backbone (note that Figure 4(b) reports the results on the ResNet-18 (He et al., 2016)). Finally, Table 1 contains evaluation metrics at test time without interventions for target and concept prediction. We observe comparable performance across methods, which are all successful due to the large dataset size and the task simplicity.

Table 1: Test-set concept and target prediction performance *without interventions*. For black boxes, concepts were predicted via a linear probe. Results are reported as averages and standard deviations across ten seeds. For concepts, performance metrics were averaged. Best results are reported in **bold**, second best are in *italics*.

| Dataset | Model | Concepts | | | Target | | |
|---|---|---|---|---|---|---|---|
| | | AUROC | AUPR | Brier | AUROC | AUPR | Brier |
| Synthetic | BLACK BOX | 0.716±0.018 | 0.710±0.017 | 0.208±0.006 | 0.686±0.043 | 0.675±0.046 | 0.460±0.003 |
| | CBM | **0.837±0.008** | **0.835±0.008** | *0.196±0.006* | **0.713±0.040** | **0.700±0.038** | *0.410±0.012* |
| | POST HOC CBM | 0.714±0.017 | 0.707±0.018 | 0.207±0.009 | 0.707±0.049 | 0.698±0.048 | **0.285±0.015** |
| | FINE-TUNED, A | — | — | — | 0.682±0.047 | 0.668±0.046 | 0.470±0.004 |
| | FINE-TUNED, MT | *0.784±0.013* | *0.780±0.014* | **0.186±0.006** | 0.687±0.046 | 0.668±0.043 | 0.471±0.003 |
| | FINE-TUNED, I | 0.716±0.018 | 0.710±0.017 | 0.208±0.006 | 0.695±0.051 | 0.685±0.051 | **0.285±0.014** |
| AwA2 | BLACK BOX | 0.991±0.002 | *0.979±0.006* | 0.027±0.006 | *0.996±0.001* | 0.926±0.020 | 0.199±0.038 |
| | CBM | *0.993±0.001* | 0.979±0.002 | *0.025±0.001* | 0.988±0.001 | 0.892±0.005 | 0.234±0.009 |
| | POST HOC CBM | 0.992±0.002 | 0.976±0.005 | *0.025±0.005* | 0.996±0.001 | *0.929±0.018* | **0.170±0.033** |
| | FINE-TUNED, A | — | — | — | 0.996±0.001 | **0.938±0.016** | **0.170±0.036** |
| | FINE-TUNED, MT | **0.994±0.002** | **0.985±0.004** | **0.022±0.005** | **0.997±0.001** | **0.938±0.017** | *0.178±0.038* |
| | FINE-TUNED, I | 0.991±0.002 | *0.979±0.005* | 0.027±0.006 | *0.996±0.001* | 0.925±0.020 | 0.195±0.040 |
| CIFAR-10 | BLACK BOX | *0.713±0.002* | *0.802±0.001* | *0.110±0.000* | *0.879±0.001* | 0.504±0.004 | 0.920±0.006 |
| | CBM | — | — | — | — | — | — |
| | POST HOC CBM | 0.675±0.009 | 0.785±0.003 | 0.125±0.004 | **0.888±0.001** | **0.541±0.004** | **0.624±0.003** |
| | FINE-TUNED, A | — | — | — | 0.876±0.002 | *0.518±0.004* | 0.896±0.005 |
| | FINE-TUNED, MT | **0.729±0.002** | **0.807±0.001** | **0.109±0.000** | 0.870±0.004 | 0.512±0.009 | *0.890±0.014* |
| | FINE-TUNED, I | *0.713±0.002* | *0.802±0.001* | *0.110±0.000* | 0.873±0.003 | 0.501±0.007 | 0.902±0.021 |
| MIMIC-CXR | BLACK BOX | 0.743±0.006 | 0.170±0.004 | *0.046±0.001* | 0.789±0.006 | 0.706±0.009 | 0.444±0.003 |
| | CBM | *0.744±0.006* | **0.224±0.003** | 0.053±0.001 | 0.765±0.007 | 0.699±0.006 | 0.427±0.003 |
| | POST HOC CBM | 0.707±0.006 | 0.154±0.006 | *0.046±0.001* | *0.801±0.006* | *0.727±0.008* | **0.301±0.005** |
| | FINE-TUNED, A | — | — | — | 0.773±0.009 | 0.665±0.013 | 0.459±0.004 |
| | FINE-TUNED, MT | **0.748±0.008** | *0.187±0.003* | **0.045±0.001** | 0.785±0.006 | 0.696±0.009 | 0.450±0.008 |
| | FINE-TUNED, I | *0.744±0.005* | 0.172±0.005 | *0.046±0.001* | **0.808±0.007** | **0.733±0.009** | *0.314±0.015* |

**Results with VLM-based Concepts**    To demonstrate that our approaches are effective *without* human-annotated concepts (Yuksekgonul et al., 2023; Oikarinen et al., 2023), we present the results on CIFAR-10 in Figure 4(c). Here, concept labels were generated using a VLM. In addition, we explore the ImageNet in Appendix F.5. The results have been obtained using the backbone architecture of Stable Difusion (Rombach et al., 2022). We do not include the CBM, as we cannot retrain such a large backbone due to computational constraints. By contrast, our method allows fine-tuning the pretrained network, thus being helpful where a CBM is impractical. As in the previous experiments, black boxes are, in principle, intervenable, and our fine-tuning approach outperforms other baselines.

**Application to Chest X-ray Classification**    To showcase the practicality of our approach, we present empirical findings on two chest X-ray datasets, MIMIC-CXR and CheXpert, primarily focusing on the former. Figure 4(d) shows intervention curves across ten independent initialisations. Interestingly, untuned black-box neural networks are not intervenable. By contrast, after fine-tuning for intervenability, the model's predictive performance and effectiveness of interventions improve visibly and even surpass those of the CBM. Given the challenging nature of these datasets, black-box model predictions may not be as strongly reliant on the considered attributes. Moreover, CBMs do not necessarily outperform black-box networks, unlike in simpler benchmarking datasets. Finally, post hoc CBMs (even with residual modelling) exhibit a behaviour similar to the synthetic dataset with incomplete concepts: interventions have only a slight positive, no, or adverse effect on performance. Analogous findings for CheXpert can be found in Appendix F.6.

## 6 Discussion & Conclusion

This work introduces a technique for performing instance-specific concept-based interventions on *any* pretrained neural network post hoc. We formalise a novel measure of *intervenability* as the effectiveness of concept-based interventions and propose a method leveraging it to fine-tune black-box models. In contrast to CBMs (Koh et al., 2020), our method circumvents the need for concept labels *during training*, which can be a challenge in practical applications. Unlike recent works on converting black boxes into CBMs post hoc (Yuksekgonul et al., 2023; Oikarinen et al., 2023), which generally do not explore *instance-specific* interventions, we propose an *effective* intervention method that is *faithful* to the original architecture and representations. Lastly, we introduce and study several other

common-sense fine-tuning baselines that perform worse than the proposed method, highlighting the need for the explicit maximisation of intervenability.

The utility of our method is highlighted empirically on synthetic tabular and natural image data. We show that, given a *small* annotated validation set, black-box models trained without explicit concept knowledge are intervenable. Moreover, our fine-tuning method improves the effectiveness of the interventions, with overall better results than alternative techniques. In addition, our approach is effective in scenarios where the concept labels are generated using VLMs. Thus, we can alleviate the need for costly human annotation while maintaining improved intervention effectiveness. Lastly, we apply the techniques in a more realistic setting of chest X-ray classification, where black boxes are not directly intervenable. The proposed fine-tuning procedure alleviates this limitation, while the other post hoc techniques are ineffective or even harmful.

**Limitations & Future Work**    Our work opens many avenues for future research and improvements. Firstly, the variant of the fine-tuning procedure considered in this paper does not affect the neural network's representations. However, it would be interesting to investigate the more general formulation wherein all model and probe parameters are fine-tuned end-to-end. According to our empirical findings, the choice of intervention strategy, hyperparameters, and probing function can influence the effectiveness of interventions. A deeper experimental investigation of these aspects is warranted. Furthermore, we only considered a single fixed intervention strategy throughout fine-tuning, whereas further improvement could come from learning an optimal strategy alongside fine-tuned weights. Beyond the current setting, we would like to apply our intervenability measure to evaluate and compare other large pretrained discriminative and also generative models.

## Acknowledgments and Disclosure of Funding

MV and SL are supported by the Swiss State Secretariat for Education, Research, and Innovation (SERI) under contract number MB22.00047.

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

# A   Motivation & Intuitive Examples

This appendix provides additional schematics and intuitive examples, clarifying instance-specific concept-based interventions. Figure A.1 schematically summarises the principle behind our intervention procedure with fewer technical details than shown in Figure 2.

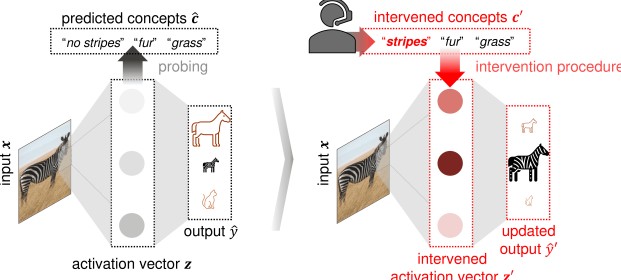

Figure A.1: Schematic summary of concept-based instance-specific interventions on a black-box neural network. This work introduces an intervention procedure that, given concept values $c'$, for an input $x$, edits the network's activation vector $z$ at an intermediate layer, replacing it with $z'$ coherent with the given concepts. The intervention results in an updated prediction $\hat{y}'$.

Figure A.2 extends on the simplified example from the main text (Figure 1). Herein, the model wrongly predicts that the image of a grizzly bear from the AwA2 dataset (Appendix D.2) depicts an otter. The user inspects the concepts via a probe and intervenes on several hand-picked common-sense variables. Our procedure updates the representations, and the predicted class is flipped to the correct one.

In addition to the model 'correction', interventions allow 'steering' the model´s prediction. In Figure A.3, the model correctly predicts a grizzly bear. The most likely prediction can be flipped to the polar bear by editing concept variables and using the proposed procedure.

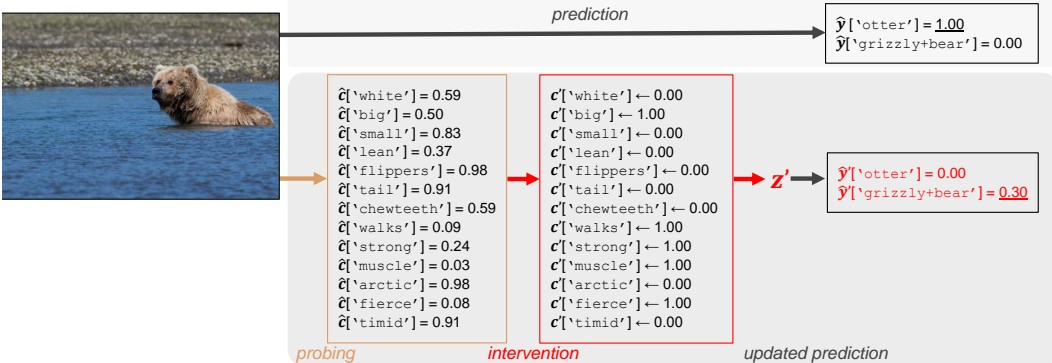

Figure A.2: A model 'correction' example using concept-based instance-specific interventions on the AwA2 dataset. The black-box neural network wrongly predicts that the image depicts an otter. Using our technique, we intervene on the network's representation and flip the final prediction to the correct class.

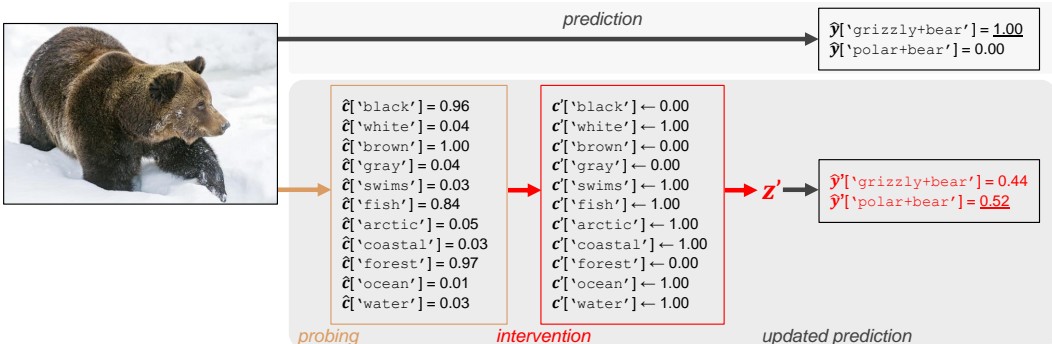

Figure A.3: A model 'steering' on the AwA2 dataset. By editing the predicted concepts and applying our method, we can manipulate the model into wrongly predicting that the image depicts a polar bear instead of a grizzly bear.

# B Fine-tuning for Intervenability

Algorithm B.1 contains the detailed pseudocode for fine-tuning for intervenability described in Section 3.3. Recall that the black-box model $f_{\boldsymbol{\theta}}$ is fine-tuned using a combination of the target prediction loss and intervenability defined in Equation 3. The implementation below applies to the special case of $\beta = 1$, which leads to the simplified loss. Importantly, in this case, the parameters $\boldsymbol{\phi}$ are treated as fixed, and the probing function $q_{\boldsymbol{\xi}}$ does not need to be fine-tuned alongside the model. Lastly, note that, in Algorithm B.1, interventions are performed for whole batches of data points $\boldsymbol{x}_b$ using the procedure described in Section 3.1.

---

**Algorithm B.1:** Fine-tuning for Intervenability

---

**Input:** Trained black-box model $f_{\boldsymbol{\theta}} = \langle g_{\boldsymbol{\psi}}, h_{\boldsymbol{\phi}} \rangle$, probing function $q_{\boldsymbol{\xi}}$, concept prediction loss function $\mathcal{L}^c$, target prediction loss function $\mathcal{L}^y$, validation set $\{(\boldsymbol{x}_i, \boldsymbol{c}_i, y_i)\}_{i=1}^N$, intervention strategy $\pi$, distance function $d$, hyperparameter value $\lambda > 0$, maximum number of steps $E_I$ for the intervention procedure, parameter for the convergence criterion $\varepsilon_I > 0$ for the intervention procedure, learning rate $\eta_I > 0$ for the intervention procedure, number of fine-tuning epochs $E$, mini-batch size $M$, learning rate $\eta > 0$

**Output:** Fine-tuned model

---

1 Train the probing function $q_{\boldsymbol{\xi}}$ on the validation set,
   *i.e.* $\boldsymbol{\xi} \leftarrow \arg\min_{\boldsymbol{\xi}'} \sum_{i=1}^N \mathcal{L}^c \left( q_{\boldsymbol{\xi}'} \left( h_{\boldsymbol{\phi}} \left( \boldsymbol{x}_i \right), \boldsymbol{c}_i \right) \right)$      ▷ Step 1: Probing

2 **for** $e = 0$ *to* $E - 1$ **do**
3      Randomly split $\{1, ..., N\}$ into mini-batches of size $M$ given by $\mathcal{B}$
4      **for** $b \in \mathcal{B}$ **do**
5          $\boldsymbol{z}_b \leftarrow h_{\boldsymbol{\phi}} \left( \boldsymbol{x}_b \right)$
6          $\hat{y}_b \leftarrow g_{\boldsymbol{\psi}} \left( \boldsymbol{z}_b \right)$
7          $\hat{\boldsymbol{c}}_b \leftarrow q_{\boldsymbol{\xi}} \left( \boldsymbol{z}_b \right)$
8          Sample $\boldsymbol{c}_b' \sim \pi$
9          Initialise $\boldsymbol{z}_b' = \boldsymbol{z}_b$, $\boldsymbol{z}_{b,\text{old}}' = \boldsymbol{z}_b + \varepsilon_I \boldsymbol{e}$, and $e_I = 0$    ▷ Step 2: Editing Representations
10          **while** $\left\| \boldsymbol{z}_b' - \boldsymbol{z}_{b,\text{old}}' \right\|_1 \geq \varepsilon_I$ **and** $e_I < E_I$ **do**
11             $\boldsymbol{z}_{b,\text{old}}' \leftarrow \boldsymbol{z}_b'$
12             $\boldsymbol{z}_b' \leftarrow \boldsymbol{z}_b' - \eta_I \nabla_{\boldsymbol{z}_b'} \left[ d(\boldsymbol{z}_b, \boldsymbol{z}_b') + \lambda \mathcal{L}^c \left( q_{\boldsymbol{\xi}} \left( \boldsymbol{z}_b' \right), \boldsymbol{c}_b' \right) \right]$    ▷ Equation 1
13             $e_I \leftarrow e_I + 1$
14          **end**
15          $\hat{y}_b' \leftarrow g_{\boldsymbol{\psi}} \left( \boldsymbol{z}_b' \right)$      ▷ Step 3: Updating Output
16          $\boldsymbol{\psi} \leftarrow \boldsymbol{\psi} - \eta \nabla_{\boldsymbol{\psi}} \mathcal{L}^y \left( \hat{y}_b', y_b \right)$      ▷ Equation 4
17      **end**
18 **end**

19 **return** $f_{\boldsymbol{\theta}}$

---

# C  Further Remarks & Discussion

This appendix contains extended remarks and discussion beyond the scope of the main text.

**Design Choices for the Intervention Procedure**  The intervention procedure entails a few design choices, including the (non)linearity of the probing function, the distance function in the objective from Equation 1, and the tradeoff between consistency and proximity determined by $\lambda$ from Equation 1. We have explored some of these choices empirically in our ablation experiments (see Figure F.4 and Appendix F). Naturally, interventions performed on black-box models using our method are meaningful in so far as the activations of the neural network are correlated with the given high-level attributes and the probing function $q_{\boldsymbol{\xi}}$ can be trained to predict these attribute values accurately. Otherwise, edited representations and updated predictions are likely to be spurious and may harm the model's performance.

**Should All Models Be Intervenable?**  Intervenability (Equation 3), in combination with the probing function, can be used to evaluate the interpretability of a black-box predictive model and help understand whether (i) learnt representations capture information about given human-understandable attributes and whether (ii) the network utilises these attributes and can be interacted with. However, a black-box model does not always need to be intervenable. For instance, when the given concept set is not predictive of the target variable, the black box trained using supervised learning should not and probably would not rely on the concepts. On the other hand, if the model's representations are nearly perfectly correlated with the attributes, providing the ground truth should not significantly impact the target prediction loss. Lastly, the model's intervenability may depend on the chosen intervention strategy, which may not always lead to the expected decrease in the loss.

**Intervenability & Variable Importance**  As mentioned in Section 3.2, intervenability (Equation 2) measures the effectiveness of interventions performed on a model by quantifying a gap between the expected target prediction loss with and without performing concept-based interventions. Equation 2 is reminiscent of the model reliance (MR) (Fisher et al., 2019) used for quantifying variable importance.

Informally, for a predictive model $f$, MR measures the importance of some feature of interest and is defined as

$$MR(f) := \frac{\text{expected loss of } f \text{ under noise}}{\text{expected loss of } f \text{ without noise}}. \tag{C.1}$$

Above, the noise augments the inputs of $f$ and must render the feature of interest uninformative of the target variable. One practical instance of the model reliance is permutation-based variable importance (Breiman, 2001; Molnar, 2022).

The intervenability measure in Equation 2 can be summarised informally as the *difference* between the expected loss of $g_{\boldsymbol{\psi}}$ without interventions and the loss under interventions. Suppose intervention strategy $\pi$ is specified so that it augments a single concept in $\hat{\boldsymbol{c}}$ with noise (Equation C.1). In that case, intervenability can be used to quantify the reliance of $g_{\boldsymbol{\psi}}$ on the concept variable of interest in $\hat{\boldsymbol{c}}$. The main difference is that Equation C.1 is given by the ratio of the expected losses, whereas intervenability looks at the difference of expectations.

**Comparison with Conceptual Counterfactual Explanations**  We can draw a relationship between the concept-based interventions (Equation 3) and conceptual counterfactual explanations (CCE) studied by Abid et al. (2022) and S. Kim et al. (2023). In brief, interventions aim to "inject" concepts $\boldsymbol{c}'$ provided by the user into the network's representation to affect and improve the downstream prediction. By contrast, CCEs seek to identify a sparse set of concept variables that could be leveraged to flip the label predicted by the classifier $f_{\boldsymbol{\theta}}$. Thus, the problem tackled in the current work is different from and complementary to CCE.

More formally, following the notation from Section 1, a conceptual counterfactual explanation (Abid et al., 2022) is given by

$$\arg\min_{\boldsymbol{w}} \mathcal{L}^y \left( g_{\boldsymbol{\psi}} \left( h_{\boldsymbol{\phi}}(\boldsymbol{x}) + \boldsymbol{w}\tilde{\boldsymbol{C}} \right), y' \right) + \alpha \|\boldsymbol{w}\|_1 + \beta \|\boldsymbol{w}\|_2 ,$$
$$\text{s.t. } \boldsymbol{w}^{\min} \leq \boldsymbol{w} \leq \boldsymbol{w}^{\max}, \tag{C.2}$$

where $\tilde{C}$ is the concept bank, $y'$ is the given target value (in classification, the opposite to the predicted $\hat{y}$), $\alpha, \beta > 0$ are penalty weights, and $\left[\boldsymbol{w}^{\min}, \boldsymbol{w}^{\max}\right]$ defines the desired range for weights $\boldsymbol{w}$. Note that further detailed constraints are imposed via the definition of $\left[\boldsymbol{w}^{\min}, \boldsymbol{w}^{\max}\right]$ in the original work by Abid et al. (2022).

Observe that the optimisation problem in Equation C.2 is defined w.r.t. the flipped label $y'$ and does not incorporate user-specified concepts $c'$ as opposed to interventions in Equation 1. Thus, CCEs aim to identify the concept variables that need to be "added" to flip the label output by the classifier. In contrast, interventions seek to perturb representations consistently with the *given* concept values.

# D    Datasets

Below, we present further details about the datasets and preprocessing involved in the experiments (Section 4). The synthetic data can be generated using our code. AwA2, CUB, CIFAR-10, ImageNet, CheXpert, and MIMIC-CXR datasets are publicly available. Table D.1 provides a brief summary.

Table D.1: Dataset summary. After any filtering or preprocessing, $N$ is the total number of data points; $p$ is the input dimensionality; and $K$ is the number of concept variables.

| Dataset | Data type | $N$ | $p$ | $K$ |
|---|---|---|---|---|
| Synthetic | Tabular | 50,000 | 1,500 | 30 |
| AwA2 | Image | 37,322 | 224×224 | 85 |
| CUB | Image | 11,788 | 224×224 | 112 |
| CIFAR-10 | Image | 60,000 | 128×128 | 143 |
| ImageNet | Image | 1,331,167 | 128×128 | 100 |
| CheXpert | Image | 49,408 | 224×224 | 13 |
| MIMIC-CXR | Image | 54,276 | 224×224 | 13 |

## D.1    Synthetic Tabular Data

As mentioned in Section 4, to perform experiments in a controlled manner, we generate synthetic nonlinear tabular data using the procedure adapted from Marcinkevičs et al. (2024). We explore two settings corresponding to different data-generating mechanisms (Figure D.1): (a) *bottleneck* and (b) *incomplete*. The first scenario directly matches the inference graph of the vanilla CBM (Koh et al., 2020). In the *incomplete* scenario, we are given incomplete concepts, *i.e.* $c$ does not fully explain the variance in $y$. Here, unexplained variance is modelled as a latent variable $r$ via the path $x \to r \to y$.

Unless mentioned otherwise, we mainly focus on the simplest scenario shown in Figure D.1(a). Below, we outline each generative process in detail. Throughout this appendix, let $N$, $p$, and $K$ denote the number of independent data points $\{(x_i, c_i, y_i)\}_{i=1}^{N}$, covariates, and concepts, respectively. Across all experiments, we set $N = 50{,}000$, $p = 1{,}500$, and $K = 30$. This dataset was divided according to the 60%-20%-20% train-validation-test split.

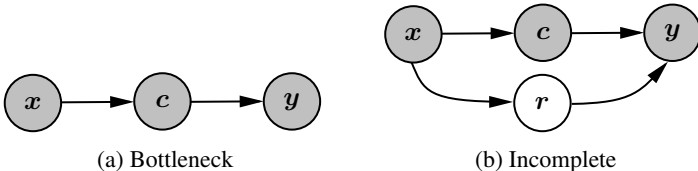

(a) Bottleneck                         (b) Incomplete

Figure D.1: Data-generating mechanisms for the synthetic dataset summarised as graphical models. Each node corresponds to a random variable. Observed variables are shown in grey.

**Bottleneck**    In this setting, the covariates $x_i$ generate binary-valued concepts $c_i \in \{0, 1\}^K$, and the binary-valued target $y_i$ depends on the covariates exclusively via the concepts. The generative process is as follows:

1. Randomly sample $\mu \in \mathbb{R}^p$ s.t. $\mu_j \sim \text{Uniform}(-5, 5)$ for $1 \leq j \leq p$.

2. Generate a random symmetric, positive-definite matrix $\Sigma \in \mathbb{R}^{p \times p}$.

3. Randomly sample a design matrix $X \in \mathbb{R}^{N \times p}$ s.t. $X_{i,:} \sim \mathcal{N}_p(\mu, \Sigma)$.[2]

4. Let $h : \mathbb{R}^p \to \mathbb{R}^K$ and $g : \mathbb{R}^K \to \mathbb{R}$ be randomly initialised multilayer perceptrons with ReLU nonlinearities.

5. Let $c_{i,k} = \mathbf{1}_{\{[h(X_{i,:})]_k \geq m_k\}}$, where $m_k = \text{median}\left(\left\{[h(X_{l,:})]_k\right\}_{l=1}^{N}\right)$, for $1 \leq i \leq N$ and $1 \leq k \leq K$.

6. Let $y_i = \mathbf{1}_{\{g(c_i) \geq m_y\}}$, where $m_y = \text{median}\left(\{g(c_i)\}_{l=1}^{N}\right)$, for $1 \leq i \leq N$.

---

[2] $X_{i,:}$ refers to the $i$-th row of the design matrix, *i.e.* the covariate vector $x_i$

**Incomplete**  Last but not least, to simulate the incomplete concept set scenario, where a part of concepts are latent, we slightly adjust the procedure from the *bottleneck* setting above:

1. Follow steps 1–3 from the *bottleneck* procedure.

2. Let $h : \mathbb{R}^p \to \mathbb{R}^{K+J}$ and $g : \mathbb{R}^{K+J} \to \mathbb{R}$ be randomly initialised multilayer perceptrons with ReLU nonlinearities, where $J$ is the number of unobserved concept variables.

3. Let $u_{i,k} = \mathbf{1}_{\left\{[h(\boldsymbol{X}_{i,:})]_k \geq m_k\right\}}$, where $m_k = \text{median}\left(\left\{[h(\boldsymbol{X}_{l,:})]_k\right\}_{l=1}^N\right)$, for $1 \leq i \leq N$ and $1 \leq k \leq K + J$.

4. Let $\boldsymbol{c}_i = \boldsymbol{u}_{i,1:K}$ and $\boldsymbol{r}_i = \boldsymbol{u}_{i,(K+1):(K+J)}$ for $1 \leq i \leq N$.

5. Let $y_i = \mathbf{1}_{\{g(\boldsymbol{u}_i) \geq m_y\}}$, where $m_y = \text{median}\left(\left\{g(\boldsymbol{u}_i)\right\}_{l=1}^N\right)$, for $1 \leq i \leq N$.

Note that, in steps 3–5 above, $\boldsymbol{u}_i$ corresponds to the concatenation of $\boldsymbol{c}_i$ and $\boldsymbol{r}_i$. Across all experiments, we set $J = 90$.

## D.2  Animals with Attributes 2

Animals with Attributes 2[3] dataset (Lampert et al., 2009; Xian et al., 2019) serves as a natural image benchmark in our experiments. It comprises 37,322 images of 50 animal classes (species), each associated with 85 binary attributes utilised as concepts. An apparent limitation of this dataset is that the concept labels are shared across whole classes, similar to the Caltech-UCSD Birds experiment from the original work by Koh et al. (2020). Thus, AwA2 offers a simplified setting for transfer learning across different classes and is designed to address attribute-based classification and zero-shot learning challenges. In our evaluation, we used all the images in the dataset without any specialised preprocessing or preselection. All images were rescaled to $224 \times 224$ pixels. This dataset was divided according to the 60%-20%-20% train-validation-test split.

## D.3  Caltech-UCSD Birds

Caltech-UCSD Birds-200-2011[4] dataset (Wah et al., 2011) is another natural image benchmark explored in the original work on CBMs by Koh et al. (2020). It consists of 11,788 bird photographs from 200 species (classes) and originally includes 312 concepts, such as wing colour, beak shape, *etc*. We have followed the preprocessing routine proposed by Koh et al. (2020) and keep the original train-validation-test split to avoid spurious mixing of photographers in the data. Particularly, the final dataset includes only the 112 most prevalent binary attributes. We have included image augmentations during training, such as random horizontal flips, adjustments of the brightness and saturation, and normalisation. Similar to AwA2, CUB concepts are shared across all instances of individual classes. No additional specialised preprocessing was performed on the images, which were rescaled to a resolution of $224 \times 224$ pixels.

## D.4  CIFAR-10

CIFAR-10[5] (Krizhevsky et al., 2009) is a benchmarking natural image dataset. It includes 60,000 $32\times32$ colour images in 10 classes, with 6,000 images per class. There are 50,000 training and 10,000 test images. To generate the validation set, we randomly hold out 10,000 images from the training data to remain faithful to the original test set. Following the setup by Oikarinen et al. (2023), we generate 143 concept labels as described in Section 4 using VLMs by comparing the similarities between each instance and the concept text embedding with thus of *not* the concept. We apply random horizontal flips, adjustments to brightness and saturation, resize the images to a resolution of $128\times128$ pixels, and apply normalisation.

---

[3]https://cvml.ista.ac.at/AwA2/
[4]https://www.vision.caltech.edu/datasets/cub_200_2011/
[5]https://www.cs.toronto.edu/~kriz/cifar.html

## D.5 ImageNet

ImageNet[6] dataset (Russakovsky et al., 2015) is a large-scale natural image benchmark. It includes 1,000 object classes and contains 1,281,167 training, 50,000 validation, and 100,000 unlabelled test images. In our experiments, we allocate half of the validation as the test set. We apply random horizontal flips, adjustments to brightness and saturation, resize images to a resolution of $128 \times 128$ pixels, and apply normalisation.

We adapt the 4,751 original concept variables introduced using GPT-3 by Oikarinen et al. (2023). To ensure that concepts are predictive of the target variable and can be inspected manually, we retain 100 attributes. Specifically, we keep those with the highest correlations with the final target while prioritising their diversity. To this end, we cluster concepts into 25 groups using the $k$-means algorithm and sample 4 attributes from each cluster based on Cramér's V (Cramér, 1999) between the concept variable and $y$. Final concept labels were generated using CLIP as for CIFAR-10(Section 4).

## D.6 Chest X-ray Datasets

As mentioned, we conducted an empirical evaluation on two real-world chest X-ray datasets: CheXpert (Irvin et al., 2019) and MIMIC-CXR (Johnson et al., 2019). The former includes over 220,000 chest radiographs from 65,240 patients at the Stanford Hospital.[7] These images are accompanied by 14 binary attributes extracted from radiologist reports using the CheXpert labeller (Irvin et al., 2019), a model trained to predict these attributes. MIMIC-CXR is another publicly available dataset containing chest radiographs in DICOM format, paired with free-text radiology reports.[8] It comprises more than 370,000 images associated with 227,835 radiographic studies conducted at the Beth Israel Deaconess Medical Center, Boston, MA, involving 65,379 patients. Similar to CheXpert, the same labeller was employed to extract the same set of 14 binary labels from the text reports. Notably, some concepts may be labelled as uncertain. Similar to Chauhan et al. (2023), we designate the *Finding/No Finding* attribute as the target variable for classification and utilise the remaining labels as concepts. In particular, the concepts are *atelectasis*, *cardiomegaly*, *consolidation*, *edema*, *enlarged cardiomediastinum*, *fracture*, *lung lesion*, *lung opacity*, *pleural effusion*, *pleural other*, *pneumonia*, *pneumothorax*, and *support devices*. In our implementation, we remove all the samples that contain uncertain labels and discard multiple visits of the same patient, keeping only the last acquired recording per subject for both datasets. All images were cropped to a square aspect ratio and rescaled to $224 \times 224$ pixels. Additionally, augmentations were applied during training, namely, random affine transformations, including rotation up to 5 degrees, translation up to 5% of the image's width and height, and shearing with a maximum angle of 5 degrees. We also include a random horizontal flip augmentation to introduce variation in the orientation of recordings within the dataset. Both chest radiograph datasets are divided according to the 80%-10%-10% train-validation-test split.

---

[6] https://www.image-net.org/update-mar-11-2021.php
[7] https://stanfordmlgroup.github.io/competitions/chexpert/
[8] https://physionet.org/content/mimic-cxr/2.0.0/

# E  Implementation Details

This section provides implementation details, such as network architectures and intervention and fine-tuning procedure hyperparameter configurations. All models and procedures were implemented using PyTorch (v 1.12.1) (Paszke et al., 2019) and scikit-learn (v 1.0.2) (Pedregosa et al., 2011). We run the reported experiments on a cluster of GeForce RTX 2080 GPUs with a single CPU worker. The span of time elapsed to run each method is dependent on the dataset and architecture. On the synthetic tabular data, on average, it takes approx. 3 hours to train a concept bottleneck or black-box model. For the Animals with Attributes 2 and chest X-ray datasets, it takes up to 10 hours to train black boxes and CBMs. However, when using a pretrained backbone, *e.g.* Stable Diffusion, only fine-tuning is required, the run-time of which ranges from 10 minutes to 1 hour for all considered datasets.

**Network & Probe Architectures**   For the synthetic tabular data, we utilise a fully connected neural network (FCNN) as the black-box model. Its architecture is summarised in Table E.1 in PyTorch-like pseudocode. For this classifier, probing functions are trained, and interventions are performed on the activations of the third layer, *i.e.* the output after line **2** in Table E.1. For AwA, CUB, MIMIC-CXR, and CheXpert, we use the ResNet-18 (He et al., 2016) with random initialisation followed by four fully connected layers and the sigmoid or softmax activation. Probing and interventions are performed on the activations of the second layer after the ResNet-18 backbone. Furthermore, we provide results for AwA with the Inception (Szegedy et al., 2015) backbone. For CIFAR-10 and ImageNet, we showcase the scalability of our methods on the pretrained Stable Diffusion Rombach et al. (2022) backbone followed by four fully connected layers and the sigmoid or softmax activation. For the CBMs, to facilitate fair comparison, we use the same architectures with the exception that the layers mentioned above were converted into bottlenecks with appropriate dimensionality and activation functions. Similar settings are used for post hoc CBMs with the addition of a linear layer mapping backbone representations to the concepts.

For fine-tuning, we utilise a single fully connected layer with an appropriate activation function as a linear probe and a multilayer perceptron with a single hidden layer as a nonlinear function. For evaluation on the test set (Table 1), we fit a logistic regression classifier from scikit-learn as a linear probe. The logistic regression is only used for evaluation purposes and not interventions.

Table E.1: Fully connected neural network architecture used as a black-box classifier in the experiments on the synthetic tabular data. `nn` stands for `torch.nn`; F stands for `torch.nn.functional`; `input_dim` corresponds to the number of input features.

| | **FCNN Classifier** |
|---|---|
| **1** | `nn.Linear(input_dim, 256)` |
| | `F.relu()` |
| | `nn.Dropout(0.05)` |
| | `nn.BatchNorm1d(256)` |
| **2** | `for l in range(2):` |
| | `    nn.Linear(256, 256)` |
| | `    F.relu()` |
| | `    nn.Dropout(0.05)` |
| | `    nn.BatchNorm1d(256)` |
| **3** | `out = nn.Linear(256, 1)` |
| **4** | `torch.sigmoid()` |

**Interventions**   Unless mentioned otherwise, interventions on black-box models were performed using linear probes, the random-subset intervention strategy, and under $\lambda = 0.8$ (Equation 1). Recall that Figures F.4 and F.2 provide ablation results on the influence of the latter hyperparameter. Despite some variability, the analysis shows that higher values of $\lambda$ expectedly lead to more effective interventions. The choice of $\lambda$ for our experiments was meant to represent the "average case", and no tuning was performed for this hyperparameter.

Similarly, we have mainly used a linear probing function and the simple random-subset intervention strategy to provide proof-of-concept results without extensive optimisation of the intervention strategy or the need for nonlinear probing. Thus, our primary focus was on demonstrating the intervenability of black-box models and showcasing the effectiveness of the fine-tuning method rather than an exhaustive hyperparameter search.

**Intervention Strategies** In ablation studies, we compare two intervention strategies (Figure F.4) inspired by Shin et al. (2023): (i) random-subset and (ii) uncertainty-based. Herein, we provide a more formal definition of these procedures described as pseudocode in Algorithms E.1–E.2. Recall that given a data point $(\boldsymbol{x}, \boldsymbol{c}, y)$ and predicted values $\hat{\boldsymbol{c}}$ and $\hat{y}$, an intervention strategy defines a distribution over intervened concept values $\boldsymbol{c}'$. Random-subset strategy (Algorithm E.1) replaces predicted values with the ground truth for several concept variables ($k$) chosen uniformly at random. By contrast, the uncertainty-based strategy (Algorithm E.2) samples concept variables to be replaced with the ground-truth values without replacement with initial probabilities proportional to the concept prediction uncertainties, denoted by $\boldsymbol{\sigma}$. In our experiments, the components of $\hat{\boldsymbol{c}}$ are the outputs of the sigmoid function, and the uncertainties are computed as $\sigma_i = 1/\left(|\hat{c}_i - 0.5| + \varepsilon\right)$ (Shin et al., 2023) for $1 \leq i \leq K$, where $\varepsilon > 0$ is small.

---

**Algorithm E.1:** Random-subset Intervention Strategy

---

**Input:** A data point $(\boldsymbol{x}, \boldsymbol{c}, y)$, predicted concept values $\hat{\boldsymbol{c}}$, the number of concept variables to be intervened on $1 \leq k \leq K$
**Output :** Intervened concept values $\boldsymbol{c}'$

1 $\boldsymbol{c}' \leftarrow \hat{\boldsymbol{c}}$
2 Sample $\mathcal{I}$ uniformly at random from $\{\mathcal{S} \subseteq \{1, \ldots, K\} : |\mathcal{S}| = k\}$
3 $\boldsymbol{c}'_{\mathcal{I}} \leftarrow \boldsymbol{c}_{\mathcal{I}}$

4 **return** $\boldsymbol{c}'$

---

**Algorithm E.2:** Uncertainty-based Intervention Strategy

---

**Input:** A data point $(\boldsymbol{x}, \boldsymbol{c}, y)$, predicted concept values $\hat{\boldsymbol{c}}$, the number of concept variables to be intervened on $1 \leq k \leq K$
**Output :** Intervened concept values $\boldsymbol{c}'$

1 $\sigma_j \leftarrow 1/\left(|\hat{c}_j - 0.5| + \varepsilon\right)$ for $1 \leq j \leq K$, where $\varepsilon > 0$ is small
2 $\boldsymbol{\sigma} \leftarrow (\sigma_1 \quad \cdots \quad \sigma_K)$
3 $\boldsymbol{c}' \leftarrow \hat{\boldsymbol{c}}$
4 Sample $k$ indices $\mathcal{I} = \{i_j\}_{j=1}^{k}$ s.t. each $i_j$ is sampled without replacement from $\{1, \ldots, K\}$
   with initial probabilities given by $(\boldsymbol{\sigma} + \varepsilon)/\left(K\varepsilon + \sum_{i=1}^{K} \sigma_i\right)$, where $\varepsilon > 0$ is small
5 $\boldsymbol{c}'_{\mathcal{I}} \leftarrow \boldsymbol{c}_{\mathcal{I}}$

6 **return** $\boldsymbol{c}'$

---

**Fine-tuning for Intervenability** The fine-tuning procedure outlined in Section 3.3 and detailed in Algorithm B.1 necessitates intervening on the representations throughout the optimisation. During fine-tuning, we utilise the random-subset intervention strategy, *i.e.* interventions are performed on a subset of the concept variables by providing the ground-truth values. More concretely, interventions are performed on 50% of the concept variables chosen uniformly at random.

**Fine-tuning Baselines** The baseline methods described in Section 4 incorporate concept information in distinct ways. On the one hand, the multitask learning approach, FINE-TUNED, MT, utilises the entire batch of concepts at each iteration during fine-tuning. For this procedure, we set $\alpha = 1.0$ (recall that $\alpha$ controls the tradeoff between the target and concept prediction loss terms). On the other hand, the FINE-TUNED, A approach, which appends the concepts to the network's activations, does not use the complete concept set for each batch. In particular, before appending, concept values are randomly masked and set to $0.5$ with a probability of $0.5$. This practical trick is reminiscent of the dropout (Srivastava et al., 2014) and is meant to help the model remain intervenable and handle missing concept values.

**Hyperparameters** Below, we list key hyperparameter configurations; the remaining details are documented in our code. For the synthetic data, CBMs and black-box classifiers are trained for 150 and 100 epochs, respectively, with a learning rate of $10^{-4}$ and a batch size of 64. Across all other experiments, CBMs are trained for 350 epochs and black-box models for 300 epochs with a learning rate of $10^{-4}$ halved midway through training and a batch size of 64. CBMs are trained using the

joint optimisation procedure (Koh et al., 2020) under $\alpha = 1.0$, where $\alpha$ controls the tradeoff between the concept and target prediction losses. All probes were trained on the validation data for 150 epochs with a learning rate of $10^{-2}$ using the stochastic gradient descent (SGD) optimiser. Finally, all fine-tuning procedures were run for 150 epochs with a learning rate of $10^{-4}$ and a batch size of 64 using the Adam optimiser. ImageNet is an exception to the above configurations due to its large size. The black-box models in this dataset were trained for 60 epochs, and the probes and fine-tuning procedures for 20 epochs. At test time, interventions were performed on batches of 512 data points.

# F  Further Results

This section contains supplementary results and ablation experiments not included in the main body of the text.

## F.1  Further Results on Synthetic Data

Figure F.1 supplements the intervention experiment results in Figure 3, Section 5, showing intervention curves w.r.t. AUPR under the *bottleneck* generative mechanism for the synthetic data with varying validation set size. The overall patterns and conclusions are similar to those observed w.r.t. AUROC (Figure 3).

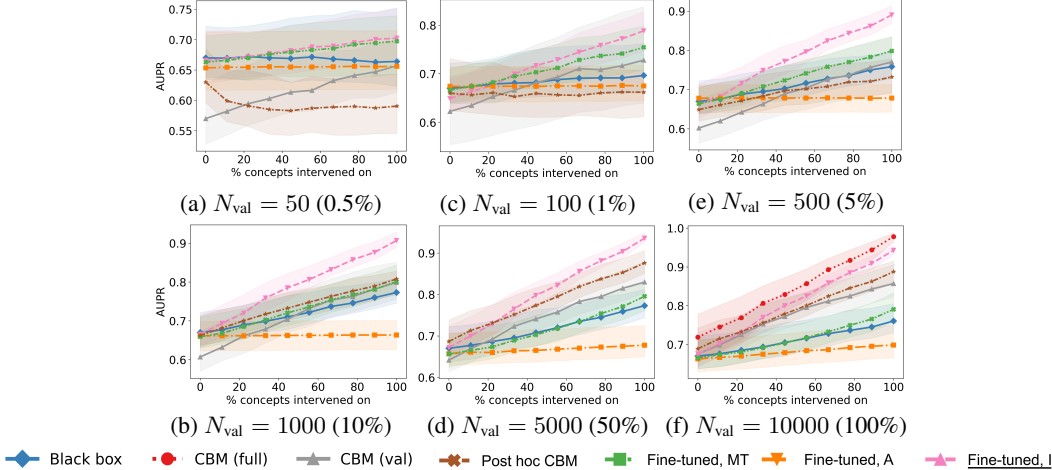

Figure F.1: Intervention results w.r.t. target AUPR on the synthetic *bottleneck* data. We explore the performance under varying validation set sizes ($N_{\text{val}}$). Percentages correspond to the fractions of the *original* validation set. For CBMs, we report the results obtained by training on the validation (**CBM val**) and full training sets (**CBM full**). Interventions were performed on test data across ten simulations. Lines correspond to medians, and confidence bands are given by interquartile ranges.

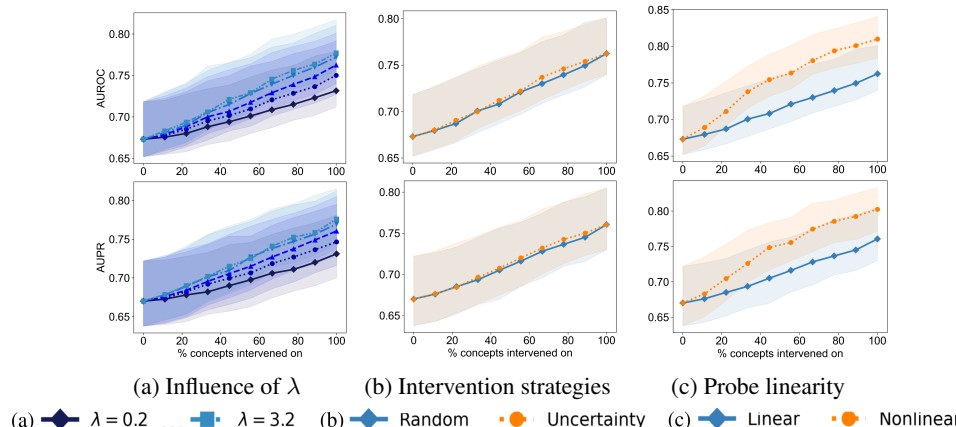

Figure F.2: Ablation study results w.r.t. target AUROC (*top*) and AUPR (*bottom*) on the synthetic dataset. Bold lines correspond to medians, and confidence bands are given by interquartile ranges across ten independent simulations. (a) Intervention results for the untuned black-box model under varying values of $\lambda \in \{0.2, 0.4, 0.8, 1.6, 3.2\}$ (Equation 3). **Darker** colours correspond to lower values. (b) Comparison between **random-subset** and **uncertainty-based** intervention strategies. (c) Comparison between **linear** and **nonlinear** probing functions.

Figure F.2 provides ablation experiment results obtained on the synthetic tabular data under the *bottleneck* generative mechanism shown in Figure D.1(a). In Figure F.2(a), we plot black-box intervention results across varying values of the hyperparameter $\lambda$ (Equation 1). Higher $\lambda$s result in more effective interventions: this finding is expected since $\lambda$ is the weight of the term penalising the inconsistency of the concept values predicted by the probe with the given values and, in the current implementation, interventions are performed using the ground truth. Interestingly, in Figure F.2(b), we observe no difference between the random subset and uncertainty-based intervention strategies. This could be explained by the fact that, in the synthetic dataset, all concepts by design are, on average, equally hard to predict and equally helpful in predicting the target variable (see Appendix D.1). Hence, the entropy-based uncertainty score should not be as informative in this dataset, and the order of intervention on the concepts should have little effect. Finally, similar to the main text, Figure F.2(c) suggests that a nonlinear probing function improves intervention effectiveness.

## F.2 Effect of Interventions on Representations

In some cases (Abid et al., 2022), it may be deemed desirable that intervened representations $z'$ (Equation 1) remain plausible, *i.e.* their distribution should be close to that of the original representations $z$. Figure F.3 shows the first two principal components (PC) obtained from a batch of original and intervened representations from the synthetic dataset (under the *bottleneck* scenario) for two different values of the $\lambda$-parameter. We observe that, under $\lambda = 0.2$ (Figure F.3(a)), interventions affect representations, but $z'$ mainly stay close to $z$ w.r.t. the two PCs. By contrast, under $\lambda = 0.4$, interventions lead to a visible distribution shift, with many vectors $z'$ lying outside of the mass of $z$. This behaviour is expected since $\lambda$ controls the tradeoff between the consistency with the given concepts $c'$ and proximity. Thus, if the plausibility of intervened representations is desirable, the parameter $\lambda$ should be tuned accordingly.

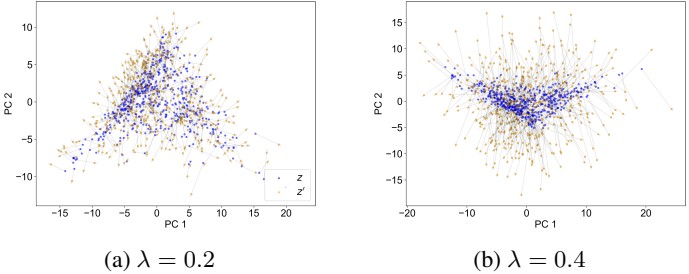

(a) $\lambda = 0.2$                     (b) $\lambda = 0.4$

Figure F.3: Principal components (PC) for a batch of data point representations **before** ($z$) and **after** ($z'$) concept-based interventions under the varying values of the parameter for (a) $\lambda = 0.2$ and (b) $\lambda = 0.4$.

## F.3 Further Results on AwA2

This section includes the ablation experiments carried out on the AwA2 dataset (Figure F.4) similar to those on the synthetic shown in Figure F.2. Firstly, we vary the $\lambda$-parameter from Equation 3, which weighs the cross-entropy term, encouraging representation consistency with the given concept values. The results in Figure F.4(a) suggest that interventions are effective across all $\lambda$s. Expectedly, higher hyperparameter values yield more effective interventions, *i.e.* a steeper increase in AUROC and AUPR. Figure F.4(b) compares two intervention strategies: randomly selecting a concept subset (random) and prioritising the most uncertain concepts (uncertainty) (Shin et al., 2023) to intervene on (Algorithms E.1 and E.2, Appendix E). The intervention strategy has a clear impact on the performance increase, with the uncertainty-based approach yielding a steeper improvement. Finally, Figure F.4(c) compares linear and nonlinear probes. Here, intervening via a nonlinear function leads to a significantly higher performance increase.

Finally, to show the scalability of our methods to different backbone architectures, we report in Figure F.5 results with the Inception (Szegedy et al., 2015) backbone. As can be seen, the intervention procedure and our fine-tuning method remain successful regardless of the backbone architectures.

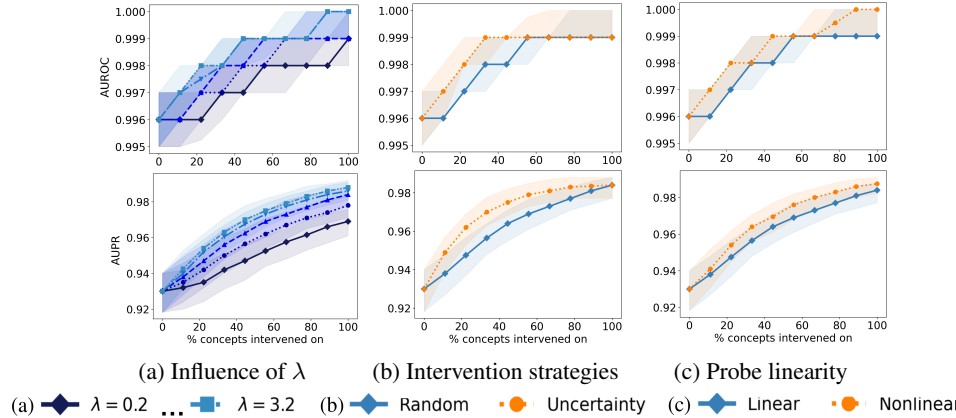

Figure F.4: Intervention results on the AwA2 dataset w.r.t. target AUROC (*top*) and AUPR (*bottom*) across ten independent train-validation-test splits. (a) Intervention results for the untuned black-box model under varying values of $\lambda \in \{0.2, 0.4, 0.8, 1.6, 3.2\}$ (Equation 3). **Darker** colours correspond to lower values. (b) Comparison between **random-subset** and **uncertainty-based** intervention strategies. (c) Comparison between **linear** and **nonlinear** probing functions.

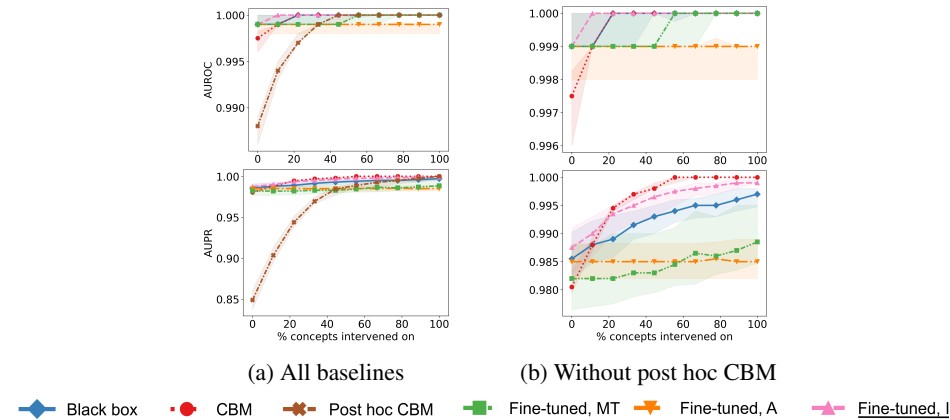

Figure F.5: Effectiveness of interventions w.r.t. target AUROC (top) and AUPR (bottom) on the AwA2 dataset with the Inception backbone. In the panels on the right (b), we have excluded post hoc CBM for legibility.

## F.4 Results on CUB

In line with the previous literature (Koh et al., 2020), we assess our approach on the CUB dataset with the results summarised in Figure F.6. This dataset is similar to the AwA2, as the concepts are shared across whole classes. Thus, concepts and classes feature a strong and simple correlation structure. Expectedly, the CBM performs very well due to its inductive bias in relying on the concept variables. As in the previous simpler scenarios, untuned black boxes are, in principle, intervenable. However, the proposed fine-tuning strategy considerably improves the effectiveness of interventions. On this dataset, the performance (without interventions) of the post hoc CBM and the model fine-tuned for intervenability is visibly lower than that of the untuned black box. We attribute this to the poor association between the concepts and the representations learnt by the black box. Interestingly, post hoc CBMs do not perform as successfully as the models fine-tuned for intervenability. Generally, the behaviour of the models on this dataset falls in line with the findings described in the main body of the text and supports our conclusions.

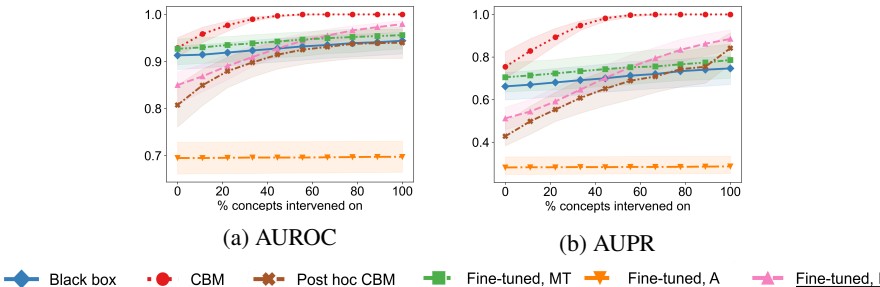

Figure F.6: Intervention results w.r.t. target (a) AUROC and (b) AUPR across ten initialisations on the CUB dataset.

## F.5 Results on ImageNet

To further support the findings on CIFAR-10 using CLIP-based concept annotations, we explore the intervention effectiveness of our method in the large-scale ImageNet dataset. In Figure F.7 we show how the proposed fine-tuning method improves the intervention effectiveness when compared with the studied baselines. Note that the CBM is not computed due to the constraint of retraining the Stable Diffusion backbone from scratch for the concept bottleneck adaptation.

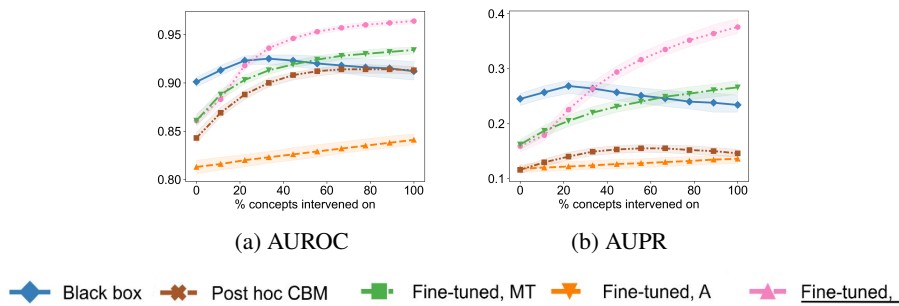

Figure F.7: Intervention results w.r.t. target (a) AUROC and (b) AUPR across ten initialisations on the ImageNet dataset.

## F.6 Results on CheXpert

To further showcase the practicality of our approach, we present empirical findings on the CheXpert dataset, which are complementary to the MIMIC-CXR results included in Section 5. Figure F.8, shows how untuned black-box neural networks are not intervenable but after fine-tuning for intervenability, the model's predictive performance and effectiveness of interventions improve visibly and even surpass those of the CBM. Finally, the remaining baseline including post hoc CBMs (even with residual modelling) exhibit a behaviour similar to the synthetic dataset with incomplete concepts: interventions have no or even an adverse effect on performance.

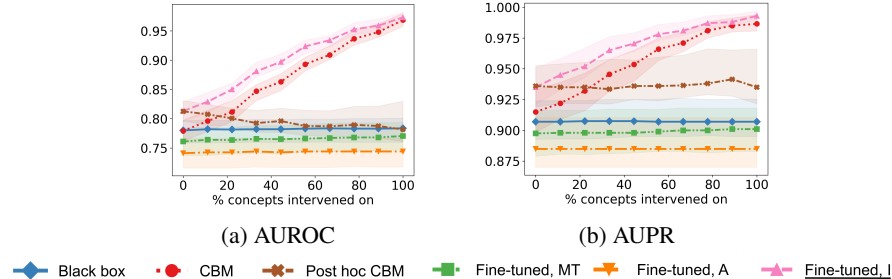

Figure F.8: Intervention results w.r.t. target (a) AUROC and (b) AUPR across ten initialisations on the CheXpert dataset.

### F.7 Calibration Results

The fine-tuning approach introduced leads to better-calibrated predictions (Table 1), possibly, due to the regularising effect of intervenability. In this section, we further support this finding by visualising calibration curves for the binary classification tasks, namely, for the synthetic tabular data and chest radiograph datasets. Figure F.9 shows calibration curves for the fine-tuned model, untuned black box, and CBM averaged across ten seeds. We have omitted fine-tuning baselines in the interest of legibility since their predictions were comparably ill-calibrated as for the black box. The fine-tuned model consistently and considerably outperforms both the untuned black box and the CBM in all three binary classification tasks, as its curve is the closest to the diagonal, which corresponds to perfect calibration.

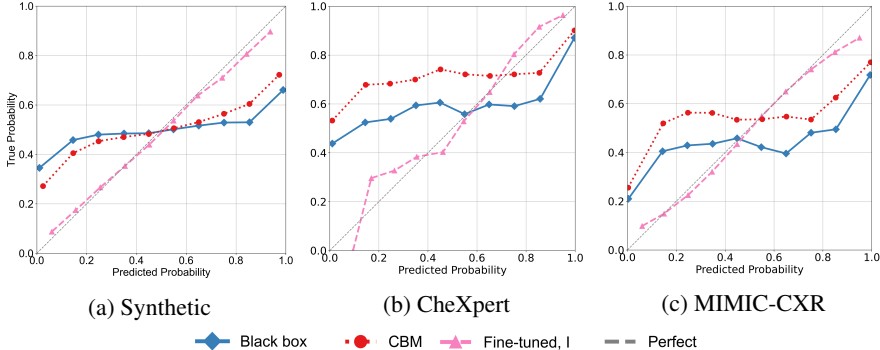

Figure F.9: Analysis of the probabilities predicted by the **black box**, **fine-tuned black box**, and **CBM** on the (a) synthetic, (b) CheXpert, and (c) MIMIC-CXR. The calibration curves, averaged across ten seeds, display for each bin the true empirical probability of $y = 1$ against the probability predicted by the model. The gray dashed line corresponds to perfectly calibrated predictions.

### F.8 Influence of the Distance Function

Throughout the experiments, we have consistently utilised the Euclidean distance as $d$ in Equation 1. In this section, we explore the influence of this design choice. In particular, we fine-tune the black-box model and intervene on all models under the cosine distance given by $d(\boldsymbol{x}, \boldsymbol{x}') = 1 - \boldsymbol{x} \cdot \boldsymbol{x}' / (\|\boldsymbol{x}\|_2 \|\boldsymbol{x}'\|_2)$.

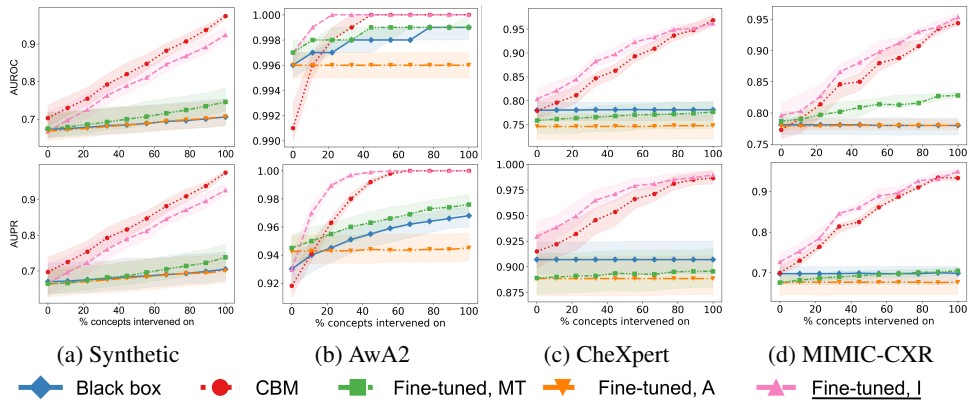

Figure F.10: Intervention results w.r.t. target AUROC (*top*) and AUPR (*bottom*) under the cosine distance function (Equation 1) on four studied datasets: (a) synthetic, (b) AwA2, (c) CheXpert, and (d) MIMIC-CXR. The comparison is performed across ten seeds.

Figure F.10 shows the intervention results under the cosine distance on the four datasets considered before. Firstly, for the synthetic and AwA2 datasets, we observe that the untuned black box is visibly less intervenable than under the Euclidean distance. In fact, for the AwA2 (Figure F.10(b), *top*), interventions slightly reduce the test-set AUROC. These results suggest that the intervention procedure is, indeed, sensitive to the choice of the distance function, and we hypothesise that the distance should be chosen to suit the latent space of the neural network considered. Encouragingly, the proposed fine-tuning procedure is equally effective under the cosine distance. Similar to the Euclidean case, it greatly improves the model's intervenability.

## F.9 Additional Baselines: Post Hoc CBMs with Residual Modelling

As an extension of post hoc CBMs, we consider residual modelling as described by Yuksekgonul et al. (2023). We refer to this baseline as POST HOC CBM-H, *i.e.* hybrid post hoc CBM. This variant adds a residual connection between the network's representations and the final output. In particular, after sequential optimisation of the post hoc CBM parameters (Section 4), a residual predictor $r_\zeta$ is trained and added to the model's output: $\min_\zeta \mathbb{E}_{(\boldsymbol{x},\boldsymbol{c},y)\sim\mathcal{D}}[\mathcal{L}^y(g_{\hat{\boldsymbol{\psi}}}(q_{\hat{\boldsymbol{\xi}}}(h_\phi(\boldsymbol{x}))) + r_\zeta(h_\phi(\boldsymbol{x})), y)]$. Similar to Yuksekgonul et al. (2023), we utilise a *linear* residual predictor in our experiments.

Figure F.11 shows intervention results for a selection of datasets. We specifically chose those experiments where simple post hoc CBMs exhibited poor intervenability. For legibility, we have only included the results for the original black box, ante and post hoc CBMs, and the model fine-tuned for intervenability. Across all datasets, POST HOC CBM-H shows a minor improvement in average predictive performance compared to the simple model. However, the introduction of the residual predictor, expectedly, has no significant effect on the steepness of the intervention curves. Thus, our fine-tuning approach consistently outperforms both variants of the post hoc CBM w.r.t. the effectiveness of interventions.

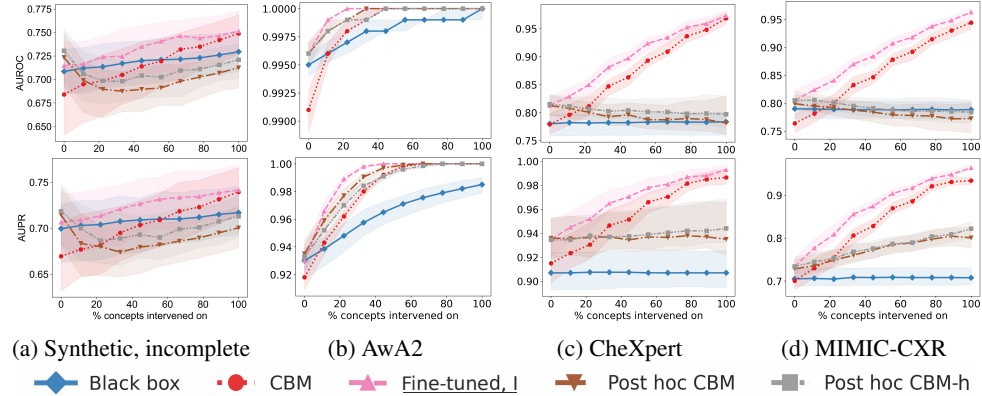

Figure F.11: Intervention results w.r.t. target AUROC (*top*) and AUPR (*bottom*) including the residual posthoc-CBM baseline on four studied datasets: (a) synthetic, (b) AwA2, (c) CheXpert, and (d) MIMIC-CXR. The comparison is performed across ten seeds.

### F.10    Influence of the CBM training method

Lastly, we explore in Figure F.12 the intervention performance of the CBMs under the three different training methods introduced by Koh et al. (2020): independent, sequential, and joint. We show in both synthetic and MIMIC-CXR datasets that the results are comparable, and therefore, throughout the manuscript, we have focused solely on the joint optimisation procedure. We chose two datasets for this ablation to span the simpler synthetic tabular scenario and the more realistic chest X-ray classification, where the concepts and final target may have a more complex dependency.

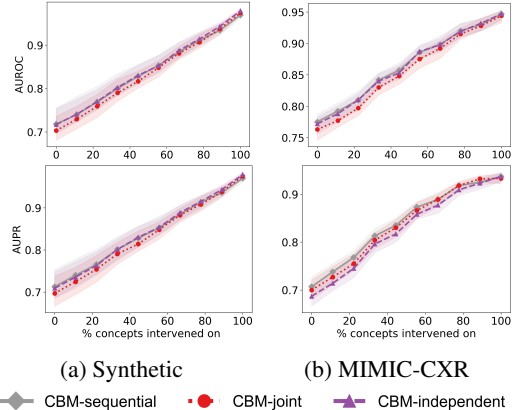

(a) Synthetic          (b) MIMIC-CXR

Figure F.12: Intervention results w.r.t. target AUROC (*top*) and AUPR (*bottom*) across ten initialisations on the (a) synthetic and (b) MIMIC-CXR datasets for the three different CBM training procedures.

