# OpenReview forum: "Beyond Concept Bottleneck Models: How to Make Black Boxes Intervenable?"
_NeurIPS.cc/2024/Conference — NeurIPS 2024 poster_

### Official Review · Reviewer_ajS6 · 2024-07-11

**Soundness:** 3
**Presentation:** 3
**Contribution:** 2
**Rating:** 7
**Confidence:** 5

**Summary:**

This work introduces a method for performing concept-based interventions on pretrained black-box neural networks, requiring only a small validation set with concept labels. The paper formalizes the notion of intervenability as a measure of the effectiveness of these interventions and experimentally assesses the intervenability of black-box classifiers across several benchmarks and architectures.

**Strengths:**

*   **Originality:** The paper presents a novel approach that enables effective concept-based interventions on black-box models. While the methods used already existed, they were creatively applied for this purpose, leading to both methodological originality and novel experimental findings.

*   **Quality:** The research methods are appropriate and well-described. The results and conclusions are supported by the data. The proposed approach is simple and sound, making it mathematically elegant and enabling the research community to use and expand it.

*   **Clarity:** The paper is well-structured with a clear flow. The writing is clear, concise, and free of grammatical errors. Figures and tables are well-designed and effectively support the text. The abstract and title summarize the paper effectively.

*   **Significance:** In their vanilla formulation, Concept Bottleneck Models were already applied on pre-trained models (e.g., ResNets) by introducing only a few extra layers on top of the backbone to predict concepts and downstream classes. Compared to vanilla CBMs, the main innovation of the proposed approach seems to consist in using concept labels only to fine-tune the model using the validation set. The findings demonstrate the flexibility of concept-based approaches and the possibility of extending concept-based interventions to black-box architectures more effectively wrt an equivalent CBM similarly fine-tuned. In view of these results, this paper could have a significant impact on real-world applications where existing pre-trained black boxes are currently used and a small concept-annotated validation set is available.

*   **Literature:** The sources are properly cited, relevant, and up-to-date.

**Weaknesses:**

*   **Clarity / Quality:** I could not find the precise training/validation procedure used for the baseline "CBM val", which represents one of the most critical baselines to compare with. I double checked both the paper but I could not find a detailed description. I also checked the code, but I could not find the configuration file (and the rest of the training/validation flow) for this baseline. The reason why I'm curious is that CBMs can also be viewed as "classification heads" fine-tuned on top of pre-trained models using a small validation set (e.g., a ResNet pre-trained on ImageNet used as a backbone of a CBM trained on CUB). The main result of this paper seems to consist in making interventions in CBMs effective with a smaller validation set. That is why the way this baseline is constructed is relevant to assess the quality and impact of this work. I could imagine at least two ways to construct such a baseline using a pre-trained backbone $g: X \to H$ (e.g., a ResNet trained on ImageNet) and a classification head $f: H \to Y$ (e.g., trained on CUB using downstream task labels but not concept labels):
 1. Fine-tune a single layer $f^{(c)}$ of $f$ to align the activations of this layer on ground-truth concept annotations of the validation set. Freeze all the other layers of the model $f$.
 2. Fine-tune all parameters before the layer $f^{(c)}$, while freezing all the layers following $f^{(c)}$.
 3. Opposite of option #2.
 4. Train a new model $f'$ using only the ground-truth concept annotations of the validation set.

 I could not understand from the paper/code to which option the baseline "CBM val" corresponded to and whether the fine-tuning approach was used to train this baseline as well.

**Questions:**

### **Major**

* **"CBM val" Baseline:** Could you clarify in detail how the baseline "CBM val" is trained and validated?

*   **“Since interventions can be executed online using a GPU (within seconds), our approach is computationally feasible”:** This is not a formal justification for why the proposed approach is computationally tractable. The time complexity can be estimated in a more formal way.


### **Minor**

*   **“This work focuses specifically”**: This paragraph is already quite technical. While clear, it could be moved to a "preliminary" section after the introduction and rephrased here to make the introduction easier for non-technical readers. Also, the notation is presented again in the methods section (which is fine as it is a technical section).

*   **Related Work:** Consider moving the related works section after the experiments. This way, you could use the experimental results to discuss the relations of your contributions with related works.

*   **Figure 2, Step 3:** The arrow in step 3 might go downwards (as in the figure in the appendix).

*   **“For a trained CBM fθ (x) = gψ (hφ (x)) = gψ (ˆ c)”**: Consider explaining again here what c^\\hat{c}c^ represents as this was mentioned in the introduction, but it was not mentioned in the presentation of the notation.

*   **“we define the intervenability as follows”**: Consider explaining the idea behind the equations (e.g., "Here, the effectiveness of interventions is quantified by" for Eq 2) before showing the equation to ease reading.

*   **“where D is the joint distribution over the covariates …”**: Consider describing the variables involved in an equation before showing the equation to avoid the reader going back and forth.

*   **“To alleviate the need for human-annotated concept values”**: This sentence might be confusing as it is not clear whether it is the proposed approach or label-free CBMs that alleviate the need for human-annotated concepts (or both). I'd rephrase it with something like "To evaluate the effectiveness of the proposed approach in the absence of human-annotated samples ..."

*   **Results section:** Organizing results by datasets leads to repetitions of some observations. I'd suggest organizing paragraphs by key findings (which might sometimes relate to a single dataset, but it is ok).

*   **Table 1:** Consider dropping "0" at the beginning of each value (e.g., ".987") or express numbers in base 100 (e.g., 98.7) to save space. Consider highlighting the best models in bold.

*   **CIFAR 10 results in Table 1:** Why not use CIFAR-100 (where concept labels are available as a ground truth) instead of CIFAR-10?


* **Discussion & Conclusion:** The largest gap with respect to baselines is on AwA and MIMIC. Do you have an intuition why this is the case? The discussion could benefit from this analysis.

**Limitations:**

Authors adequately discuss limitations and future works in a dedicated section.

---

> ### Author Rebuttal · Authors · 2024-08-06
>
> Thank you for the feedback! You will find our point-by-point responses below.
> > I could not find the precise training/validation procedure used for the baseline "CBM val".
>
> We have updated the public repository to include the configuration file `config_synthetic_cbm_val.yaml` and code to run the “CBM val” baseline. As mentioned, CBMs can be viewed as classification heads on top of pre-trained models. However, this would correspond to the “post hoc CBM”. We frame the presented CBM in its original “ante hoc” implementation [6], where the backbone is trained from scratch. Our work focuses on making interventions on *any pre-trained black box* more effective, not the CBMs. From the options provided by the reviewer, our “CBM val” is number *4*. This way, we show the sample efficiency of our method over the (ante hoc) CBMs when only a small portion of annotated data is available.
> >“Since interventions can be executed online using a GPU, our approach is computationally feasible”: This is not a formal justification for why the proposed approach is computationally tractable.
>
> This statement is not meant as a formal complexity metric, but as a remark for the interested reader that the interventions are less costly than training or fine-tuning and can be run “online” with a regular GPU. As the intervention procedure is gradient-based, the computational cost varies across datasets and depends on the optimization iterations. The number required depends on several factors, e.g., the value of $\lambda$, the number of concepts, the dimensionality of $z$, and the number of data points per batch. Given these, providing a formal description of the running-time complexity is challenging.
> For example, on the synthetic dataset, with the convergence parameter set to $10^{-6}$ (very conservative) and for a batch of 512 samples, the intervention procedure requires approx. 500 to 2500 steps (the number varies depending on the number of given concepts and the value of $\lambda$), which amounts to 0.5 to 2 s (for the whole batch) for an untuned black-box model. We use smaller batches and more permissive convergence criteria when fine-tuning, allowing a considerable speedup. In addition, note that the run-time of the intervention procedure is not strictly dependent on the black-box model’s architecture (except for the dimensionality of the layer intervened on).
> > “This work focuses specifically”: This paragraph is already quite technical. While clear, it could be moved to a "preliminary" section after the introduction.
>
> Thank you for the suggestion! In order to address a broader audience, we will include a preliminary section introducing the method with the technical contributions, and reframe the introduction to be less technically oriented, removing the formal notation.
> > Figure 2, The arrow in step 3 might go downwards.
>
> We will update the figure.
> > “For a trained CBM fθ (x) = gψ (hφ (x)) = gψ (ˆ c)”: Consider explaining again here what c^\hat{c}c^ represents
>
> With the new organization of the introduction and preliminary section of the methods, we will introduce the notation accordingly prior to the description of the method.
> > “we define the intervenability as follows”: Consider explaining the idea behind the equations before showing the equation to ease reading.
>
> Intuitively, the more effective an intervention is, ideally, the more it should reduce the target prediction error. Leveraging this, our equation measures the gap between the prediction error prior and post intervention, where larger gaps correspond to more effective interventions.
> > **I**. “where D is the joint distribution over the covariates …”: Consider describing the variables involved in an equation before showing the equation to avoid the reader going back and forth. **II**. “To alleviate the need for human-annotated concept values”: This sentence might be confusing as it is not clear, I'd rephrase it. **III**. Table 1: Consider dropping "0" at the beginning of each value or express numbers in base 100 to save space.
>
> We will clarify and introduce the notation in **I**, rephrase **II**, and change the metrics to percentages and bold the best results in **III**.
>
> > CIFAR 10 results in Table 1: Why not use CIFAR-100 (where concept labels are available as a ground truth) instead of CIFAR-10?
>
> We include experiments with both CIFAR-10 and ImageNet that do not originally have available ground-truth concepts to show that our method can be combined with “concept discovery” [7], where annotations are acquired through VLMs. We did not include CIFAR-100 as we have a synthetic tabular, two natural image datasets (CUB and AwA2), and two medical image datasets (MIMIC-CXR and CheXpert) that, we believe, are sufficiently representative to cover the settings with “annotated” concepts.
> > The largest gap with respect to baselines is on AwA and MIMIC. Do you have an intuition why this is the case?
>
> Generally, our procedure has a stronger effect because we target the improvement with interventions with a stronger inductive bias, while other methods do not explicitly encourage intervention effectiveness. In some instances, activations are correlated with concepts, but the network does not rely on them for prediction, e.g., when there is a complex relation between the concepts and target, as in the case of X-rays (MIMIC), and only probing is not enough to intervene effectively. However, after fine-tuning, the black box learns to use the information correlated with the concepts at prediction time.  At the same time, our approach does not require introducing a concept bottleneck layer, and, therefore, the model may still use additional non-concept-based information from the representations. With respect to the AwA (and CIFAR-10), the experimental findings fall within our expectations: in both, concepts are useful for predicting the target variable, and our fine-tuning explicitly encourages intervention effectiveness, providing a stronger supervisory signal.

---

> > ### Comment · Reviewer_ajS6 · 2024-08-08
> >
> > Thank you for your effort and for taking the time to explain your work even further!
> >
> > I am aware that the following comments/questions are not critical nor the focus of this work. I am asking out of curiosity.
> >
> > **[nit] Computational cost**: Thank you for providing a more detailed discussion on this. I think the paper would benefit if you could report this discussion in the appendix and take it a bit further by ranking the most critical factors and estimate their relationship with the computational cost (e.g., linear, quadratic, etc).
> >
> > **[nit] CIFAR**: I understand the motivation for choosing CIFAR without using concept annotations. I'm wondering though whether the concept annotations generated using the VLM are aligned in some way to the original concept labels in CIFAR-100. And in case VLM and CIFAR-100 concept labels overlap, I'm curious to understand whether: (i) the VLM's annotations are actually matching the ground truth, and (ii) the interventions would be more effective using VLM's annotations or ground-truth annotations.

---

> ### Author Response · Authors · 2024-08-09
> **Answer to Official Comment Reviewer ajS6**
>
> Thank you for your nice feedback and for your interest in our work!
>
> As a follow up to the two raised points:
>
> - **Computational cost**: We agree with your point and will include the discussion in the appendix, thank you! Regarding the most critical factors, we believe all, the convergence parameter, number of concepts, number of datapoints, value of lambda, and the dimensionality of z play a relevant role. Due to the nature of the gradient-based optimization that runs until convergence, it is challenging to provide a formal estimation of the cost or ranking. However, one could try to approach the problem empirically by providing wall-time results of the optimization under controlled synthetic experiments varying the mentioned parameters to provide a better understanding of their relevance, which we will include in the appendix of the revised paper. We will, in addition, consider characterising more formally the complexity of a *single* update step in this gradient-based procedure, which should give additional intuition on the complexity.
>
>
> - **CIFAR**: We believe that the original concepts coming from CIFAR-100, i.e. the classes, and those from [7]  in the VLM scenario differ significantly. More precisely, the original work in [7] introduces a total of 824 concepts in CIFAR-100 while the original annotated labels are only 100, and cover different categories. We assume in this discussion that the “concepts” the reviewer refers to in CIFAR-100 are the super-classes. For this reason, using the VLM-based concepts to intervene on a model trained with the original CIFAR-100 concepts would not be possible. However, we envision a setup where keeping the original concept categories, the CIFAR-100 dataset can be re-annotated using CLIP, this way the experiment proposed by the reviewer could be conducted. Based on our experience with VLM annotated data, we would expect it to have more noise and, therefore, potentially, less effective interventions than under the ground-truth annotations.

---

> > ### Comment · Reviewer_ajS6 · 2024-08-12
> >
> > I agree on both points!
> > Congrats again for your work!

---

### Official Review · Reviewer_Q4Tj · 2024-07-12

**Soundness:** 4
**Presentation:** 2
**Contribution:** 4
**Rating:** 7
**Confidence:** 5

**Summary:**

The paper proposes a novel method for test-time concept-based intervention on black-box models. Starting from the interactive intervention procedure of concept-bottleneck models, the authors devise a novel technique for reproducing it on any pre-trained black-box model. It consists of training a concept probe on the latent space of the model and using it to modify the representation of a given concept and checking the response of the model to this intervention. Also, the authors define a measure of intervenability which can be optimized to fine-tune a model architecture, while making it more prone to concept-intervention. The proposed approach is tested over several benchmarks.

**Strengths:**

**Novelty and impact**: although based on the idea of concept probing (step 1, already proposed in literature), the following steps of editing the representations and updating the output (i.e., of performing an intervention) of a black-box model are simple but very interesting and potentially important for a large part of the XAI community. Indeed, it brings for the first time the interactability of explainable-by-design concept-based models to post-hoc concept-based techniques.

**Weaknesses:**

## Major issues
- **Presentation**: the paper is not very clear in a few passages. Also, some comments to the results in the experimental sections are excessive or not completely true, and they must be changed/softened. See minor issues for indications of the individual sentences to edit.
- **Equation 4**: presenting Eq. 4 and then only considering the special case of $\beta=1$ is not correct, as it completely neglects a term. This is not acceptable even if it implies the training of the probe, a reader expects to see an ablation study on that in the experiment. I solicit the authors to present it otherwise (e.g., presenting first the actual optimized equation) or provide an ablation study on that.

## Minor issues
- In the abstract, “Recently, interpretable machine learning has re-explored concept bottleneck models (CBM)”. The sentence is not very clear, it seems in this way that concept bottleneck models have already been proposed in the past. You also say it in the introduction. If it is the case, and it has been proposed earlier, you should provide a citation.
- “While, in principle, a specialist may just override predictions, in many cases, concept and target variables are linked via nonlinear relationships potentially unknown to the user.” This sentence is not super clear and does not seem to support the paper point. I would consider rephrasing or removing it.
- Figure 1: The concepts reported are not clear. Another example or the same example with textual concepts instead of icons would be probably easier to read.
- In the Method, “we will assume being given a small validation set”, I would explicitly add “equipped with concept annotations”.
- “think of a doctor trying to interact with a chest X-ray classifier $(f_\theta)$ by annotating their findings”, which findings? Not very clear, why don’t you directly provide an example of concept for the X-ray task?
- “Note that, herein, an essential design choice explored in our experiments is the (non)linearity of the probe”. Not clear from this whether you consider both a linear and a non-linear, or only one of the two without looking at the experiments.
- You should probably cite also TCAV [1] in 3.1 when talking about probing. The CAV is probably the most renowned concept probe.
- I think you should better explain and point out the fact that if you directly optimize equation 3 you could decrease the performance of the model, since you would maximize the error over the non-intervened network. Instead, you says “ Equation 3 is differentiable, a neural network can be fine-tuned by explicitly maximising”. And, in equation 4, you change the optimization problem optimizing both losses
- Fine-tuned, A: it is not very clear the reason of including this model, nor the explanation. Please rewrite and add further motivations.
- Commenting Figures F.4 and Figure 4 this close is confusing. If possible, position the appendix figures in a different order to avoid confusing the reader.
- “Finally, post hoc CBMs (even with residual modelling) exhibit a behaviour similar to the synthetic dataset with incomplete concepts: interventions have no or even an adverse effect on performance.” This claim is not supported by the results presented in figure 4(d) bottom: the AUPR is increasing in a non-negligible manner. Please rephrase the sentence. A similar consideration holds also for the Discussion section.

[1] Kim, Been, et al. "Interpretability beyond feature attribution: Quantitative testing with concept activation vectors (tcav)." International conference on machine learning. PMLR, 2018.

**Questions:**

1. Table 1 report results where the black-box model is normally not that good for the target performance (among the worst results in general). This is something that I would not have expected, since both the bottleneck and the fine-tuning pose further constraints that normally impact the model performance (as also reported by CBM and many other concept-based approaches). How do you explain it?
2. Why is the CBM more computationally expensive in the experiments with VLM-based Concept? I think you could use a fixed CLIP to predict the concepts and simply train the task predictor on that. If I am right, I think the following sentence is unfair, and I suggest removing it “By contrast, our method allows fine-tuning the pretrained network, thus being helpful where a CBM is impractical”.
3. In Cher X-ray classification the black-box models are not intervenable. This is a surprising result, which partly contrast with the claim that you propose a method to make “interventions on any pretrained neural network post hoc. Could you elaborate further on that?

**Limitations:**

The authors report mostly possible expansions of their work, rather than actual limitations and possible societal implications of their approach. Indeed, even though the proposed method improves the interpretability of the model, it may impact its security. An attacker, for instance, could misuse the proposed technique for detecting model biases towards some specific concepts and creating realistic adversarial attacks that could go undetected.

---

> ### Author Rebuttal · Authors · 2024-08-06
>
> Thank you for your detailed comments! Below, we respond to your concerns point by point.
> > Equation 4: presenting Eq. 4 and then only considering the special case of β=1 is not correct.
>
> We believe the formalization of the overall optimization problem is a beneficial contribution to future lines of work. However, we agree our focus is on the simplified case and will adapt the manuscript first introducing the special case, followed by the general formulation.
> >In the abstract, “Recently, interpretable machine learning has re-explored CBMs”. The sentence is not very clear, it seems in this way that CBMs have already been proposed in the past.
>
> We refer to the works referenced in “Introduction” and “Related Work” [4,5] and will make them more explicit. They propose concept-based approaches to classification. Although not strictly referred to as CBMs, they are very similar and the original work on CBMs [6] acknowledges them as closely related.
> > “While a specialist may just override predictions, concept and target variables are linked via nonlinear relationships potentially unknown to the user.” This sentence is not super clear and does not seem to support the paper point.
>
> We motivated the need for concept-based interventions, especially in scenarios where a professional has the concept information but inferring the target variable is not an easy task. We understand the potential confusion and will remove it.
> > Figure 1: The concepts reported are not clear.
>
> We will include their description in the main text and caption. Particularly, in order of appearance from left to right, and top to bottom, the meaning of the icons is: “fierce”, “timid”, “muscle”, “walks”, “otter” and “grizzly bear”. This figure is complemented by a more concrete example of model correction in Figure A.2 with a list of all intervened on concepts.
> > “think of a doctor trying to interact with a chest X-ray classifier by annotating their findings”, which findings?
>
> The findings are the concepts of the X-ray datasets. We will provide examples to help understand the X-ray use-case and include a table in the appendix with the full list of concepts: atelectasis, cardiomegaly, consolidation, edema, enlarged cardiomediastinum, fracture, lung lesion, lung opacity, pleural effusion, pleural other, pneumonia, pneumothorax, and support devices.
> > Not clear whether you consider both a linear and a non-linear probe, or only one of the two without looking at the experiments.
>
> We will clarify that the experiments shown in the main body are performed under a linear probe and an ablation with a nonlinear probe is found in the appendix.
> > You should probably cite also TCAV in 3.1 when talking about probing. The CAV is probably the most renowned concept probe.
>
> We will include it.
> > I think you should better explain the fact that if you directly optimize equation 3 you could decrease the performance of the model.
>
> We will make it more explicit that the final loss function is a linear combination of the intervenability measure in Eq. 3 and the target prediction loss, resulting in Eq. 4. We always consider the combination of the two to better control the tradeoff.
> > Fine-tuned, A: explanation to why including this model
>
> The motivation is: it would be natural to include the concepts in the model, either at the input or at intermediate layers. Since the considered feature spaces are high-dimensional, we append concept variables to the intermediate activations as a direct comparison to Fine-tuned I.
> > “Finally, post hoc CBMs exhibit this behaviour: interventions have no or even an adverse effect on performance.” This claim is not supported by the results presented in figure 4(d) bottom.
>
> We will rephrase the statement to clarify that in some datasets, a small positive effect is present.
> > Table 1 report results where the black-box model is normally not that good for the target performance.
>
> In synthetic data, the generating process favors the CBM architecture, as concepts directly relate the inputs to the target and we expect the CBM to outperform. In the remaining datasets with more complex concept-to-target dependencies, the CBM performs worse, as expected. If the concept variables are “useful” for the target prediction, we hypothesise the fine-tuning methods would improve predictive performance.
>
> Specifically, fine-tuning enforces reliance on the concept variables, which might increase the robustness and prevent overfitting. Moreover, fine-tuned black boxes keep the original representation without introducing a bottleneck layer and can still use non-concept-based information if necessary. To finalize, our focus was on making the models more intervenable, and Table 1 suggests that most methods behave on par, i.e., fine-tuning with intervenability does not harm the performance.
> > Why is the CBM more computationally expensive in the experiments with VLM-based Concept? I think you could use a fixed CLIP to predict the concepts.
>
> The VLM-based concept annotation does not influence the computational cost of the CBMs. We have used these annotations to explore larger backbone architectures and training these from scratch, as would be needed for the (ante hoc) CBM baseline, is not computationally feasible. It would require a large number of concept annotations and resources. Using a fixed CLIP to predict the concepts is alike the post hoc CBM, which we include as baseline.
> > In Cher X-ray classification the black-box models are not intervenable. This partly contrast with the claim that you propose a method to make “interventions on any pretrained neural network post hoc.
>
> In some instances, activations are correlated with concepts, but the network does not necessarily rely on them for prediction, hence, only probing is not enough to intervene effectively. However, after fine-tuning, the black box learns to use the information correlated with the concepts at prediction time. Hereafter, fine-tuning may be necessary in some instances.

---

> > ### Comment · Reviewer_Q4Tj · 2024-08-12
> >
> > I thank the authors for taking the time to fix the issues and answer my questions. I think the paper is ready to be accepted now.

---

### Official Review · Reviewer_W2n5 · 2024-07-12

**Soundness:** 3
**Presentation:** 4
**Contribution:** 3
**Rating:** 7
**Confidence:** 4

**Summary:**

The paper proposes a method to make any pre-trained black box model intervenable, given a small validation dataset with concept.

This is done by the following procedure:
- Extract the output of a layer of a black-box $g_{	\psi} (z)$ and train a probing network $q_{\xi}(z)$ to extract the concepts from that layer using the validation dataset.
- For intervening i.e changing concepts $c$ to $c'$: Given $z$ find $z'$ that minimizes $L^c(q_{\xi}(z'),c')+d(z,z')$ where $d$ is a distance measure. $z'$ is optimized using gradient descent.
- Train the black-box network to be intervenable by finetuning, minimize $L^y(g_{\psi}(z'),y)$
- To edit the representations,  replace $z'$ with $z$.

The paper demonstrated the effectiveness of their approach by comparing it to the following baselines:
- A black-box with representation editing and no finetuning.
- CBM
- A finetune black-box that predicts the concepts and the output (Multitask-learning)
- A finetune black-box by appending the black box activations with the concepts.
- Post-hoc CBMs

The approach was evaluated on the following datasets:
- A synthetic tabular dataset with a complete and incomplete concept dataset.
- Vision datasets with labels: AWA2 and CUB
- Vision datasets with no labels: Cifar10 and imageNet (Here labels are obtained from an LLM in a label-free style).

Results that the proposed method outperforms others in cases where data is limited i.e the % of validation dataset is low or when the number of interventions is generally low.

**Strengths:**

### Originality -- Good

- The method itself is original, i.e., how the representations are found and training is done is novel. While there are post-hoc CBMs that can allow intervention on the black box as well. The proposed method seems to outperform them, especially in the incomplete data regime.

### Quality-- Excellent

- The proposed method is simple but seems very effective and the fine-tuning procedure is well motivated by an empirical comparison to black-box model interventions without the fine-tuning.

- Very good empirical evaluation in general, comparisons were done on synthetic data, and 4 different image data sets on different domains; with reasonable baselines.

- Code was provided to reproduce the experiments.

### Clarity -- Excellent
- The paper is well-written, clear and easy to follow.

**Weaknesses:**

### Significance -- Fair

- There are two main advantages of using CBMs, interpretability and intervenability. While the proposed method allows for intervenability, this method does not add interpretability to black box model.

- One could argue that since you need a validation set with concept anyhow, one should just put a CBM in the original architecture giving both interpretability and intervenability. Empirical validation in the paper did show that the proposed method did better than CBM however baseline CBM used in the evaluation was not trained for intervenability as done by Mateo, et al. [1] which could dramatically improve the performance of regular CBMs. So its difficult to conclude that the proposed method is in fact superior to CBMs in any way.



[1] Espinosa Zarlenga, Mateo, et al. "Learning to receive help: Intervention-aware concept embedding models." Advances in Neural Information Processing Systems 36 (2023).

**Questions:**

- Which layer of is used for interventions, does it matter?
- For the multi-task baseline, Fine-Tune MT how do you intervene on test time since changing the multitask label does not affect the output of the classifier?
- For the appending baseline Fine-Tune A do you throw away the last layer of the original network (i.e the layer after the one used for interventions) and create a new one to handle the new size?

**Limitations:**

Yes.

---

> ### Author Rebuttal · Authors · 2024-08-06
>
> We thank the reviewer for the feedback and questions! Below is our point-by-point response.
>
> > There are two main advantages of using CBMs, interpretability and intervenability. While the proposed method allows for intervenability, this method does not add interpretability to black box model.
>
> As stated by the reviewer, this work focuses on intervenability and its formalisation. However, the proposed method also incorporates interpretability with respect to a regular black box model. By mapping intermediate layers to human-understandable concepts via the probe, the user gains a better interpretation of the information contained in the representations. This connection between probing classifiers and interpretability is supported and further detailed in prior work referenced in the manuscript [1,2,3].
>
> > One could argue that since you need a validation set with concept anyhow, one should just put a CBM in the original architecture giving both interpretability and intervenability. Empirical validation in the paper did show that the proposed method did better than CBM however baseline CBM used in the evaluation was not trained for intervenability as done by Mateo, et al. [1] which could dramatically improve the performance of regular CBMs. So its difficult to conclude that the proposed method is in fact superior to CBMs in any way.
>
> The original CBM architecture requires training the concept and target predictors from scratch. This is a limitation in the settings where there is not enough data available to train a large backbone, e.g. using Stable Diffusion like in the CIFAR-10 and ImageNet experiments. This problem would be aggravated further if only a validation set with concept annotations was used. We explore this in Figure 3 and show that CBMs are not sample efficient when trained only on the validation set.
>
> We did a preliminary exploration of the introduction of interventions at training time in the CBM, similar to the work done on the mentioned CEMs by Mateo et al., and we did not find a significant improvement with respect to the regular CBM in the presented datasets.
>
> Effectively, intervening on a vanilla CBM at training time can be thought of as a combination of independent and joint training, which, as shown in Appendix F.10, have similar results.
>
> In the cases where the concept set is incomplete, the bottleneck layer in the CBM will hinder the overall model performance, even under interventions. However, finetuning for intervenability does not limit the expressivity of the intervened-on layer and, therefore, allows for better target prediction.
>
> >Which layer of is used for interventions, does it matter?
>
> The reported experiments show the results of intervening two non-linear layers after the backbone. The layer you intervene on will play a role in the effectiveness of interventions. Generally, high-level information in neural networks emerges in deeper layers, which is why we recommend conducting the interventions there. Additionally, the required complexity of the probe is lower as the dimension of the layers is reduced deeper within the network. Prior works show probing at deeper layers is benefitial [2]. However, the proposed method can be applied at any step, provided the layer has extracted enough information, and we expect interventions to have a positive effect on the target prediction.
>
> >For the multi-task baseline, Fine-Tune MT how do you intervene on test time since changing the multitask label does not affect the output of the classifier?
>
> The intervention on Fine-Tune MT is done in the same fashion as in the proposed Fine-Tune I, using the three-step solution introduced in Section 3.1. in the manuscript. This will be made clarer in the revised version of the paper.
>
> >For the appending baseline Fine-Tune A do you throw away the last layer of the original network (i.e the layer after the one used for interventions) and create a new one to handle the new size?
>
> This is correct. We leverage the pretrained backbone until the layer where the concatenation with the concepts occurs, in this case, two non-linear layers after the extracted features. At this state, a new target predictor is created with input layer size correspondig to the representations concatenated with the concepts.

---

> > ### Comment · Reviewer_W2n5 · 2024-08-12
> > **Thanks**
> >
> > I would like to thank the authors for their response, very interesting work, hope to see this paper in the conference.

---

### Author Rebuttal · Authors · 2024-08-06

Dear reviewers,

We thank you for the detailed feedback; we will make sure to address the concerns and incorporate the corresponding changes in the revised manuscript upon acceptance. Below is a summary of the main responses and clarifications addressed in this rebuttal:
- We have provided a more in-depth explanation of the baseline “CBM val”. Effectively, it consists of training an ante hoc CBM solely on the validation set to compare CBM’s sample efficiency against the proposed method.
- In our experiments, we have compared with two baselines related to CBMs, namely, (ante hoc) CBM and post hoc CBM. We would like to emphasise that the (ante hoc) CBMs represent the original formulation of CBMs, i.e. *without* pretrained backbones. We will clarify the distinction between the (ante hoc) CBM and post hoc CBM used as baselines in the revised manuscript.
- Our method focuses on the intervenability of the models. However, it also facilitates interpretability via representation probing.

*References*:

[1] Belinkov, Y. (2022). Probing Classifiers: Promises, Shortcomings, and Advances. Computational Linguistics, 48(1), 207-219.

[2] Alain, G., & Bengio, Y. (2016). Understanding intermediate layers using linear classifier probes. arXiv preprint arXiv:1610.01644.

[3] Kim, Been, et al. (2018) Interpretability beyond feature attribution: Quantitative testing with concept activation vectors (TCAV). International conference on machine learning. PMLR.

[4] Lampert, C. H., Nickisch, H., & Harmeling, S. (2009). Learning to detect unseen object classes by between-class attribute transfer. In 2009 IEEE conference on computer vision and pattern recognition. Miami, FL, USA: IEEE.

[5] Kumar, N., Berg, A. C., Belhumeur, P. N., & Nayar, S. K. (2009). Attribute and simile classifiers for face verification. In 2009 ieee 12th international conference on computer vision (pp. 365–372). Kyoto, Japan: IEEE.

[6] Koh, P. W., Nguyen, T., Tang, Y. S., Mussmann, S., Pierson, E., Kim, B., & Liang, P. (2020). Concept bottleneck models. In H. D. III & A. Singh (Eds.), Proceedings of the 37th international conference on machine learning (Vol. 119, pp. 5338–5348). Virtual: PMLR.

[7] Oikarinen, T., Das, S., Nguyen, L. M., & Weng, T.-W. (2023). Label-free concept bottleneck models. In The 11th international conference on learning representations.

---

### Decision · Program_Chairs · 2024-09-25

**Decision:**

Accept (poster)

**Comment:**

All reviewers find the proposed approach to make a pre-trained black box model interpretable very interesting.
All reviewers clearly vote for accepting this paper.